# 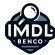 IMDL-BenCo: A Comprehensive Benchmark and Codebase for Image Manipulation Detection & Localization

**Xiaochen Ma**[1†], **Xuekang Zhu**[1†], **Lei Su**[1†], **Bo Du**[1†], **Zhuohang Jiang**[1†], **Bingkui Tong**[1†], **Zeyu Lei**[1,2†], **Xinyu Yang**[1†], **Chi-Man Pun**[2], **Jiancheng Lv**[1,3], **Jizhe Zhou**[1,3*]

[1]College of Computer Science, Sichuan University, China
[2]Department of Computer and Information Science, University of Macao, Macao SAR
[3]Engineering Research Center of Machine Learning and Industry Intelligence, MOE, China

## Abstract

A comprehensive benchmark is yet to be established in the Image Manipulation Detection & Localization (IMDL) field. The absence of such a benchmark leads to insufficient and misleading model evaluations, severely undermining the development of this field. However, the scarcity of open-sourced baseline models and inconsistent training and evaluation protocols make conducting rigorous experiments and faithful comparisons among IMDL models challenging. To address these challenges, we introduce IMDL-BenCo, the first comprehensive IMDL benchmark and modular codebase. IMDL-BenCo: **i)** decomposes the IMDL framework into standardized, reusable components and revises the model construction pipeline, improving coding efficiency and customization flexibility; **ii)** fully implements or incorporates training code for state-of-the-art models to establish a comprehensive IMDL benchmark; and **iii)** conducts deep analysis based on the established benchmark and codebase, offering new insights into IMDL model architecture, dataset characteristics, and evaluation standards. Specifically, IMDL-BenCo includes common processing algorithms, 8 state-of-the-art IMDL models (1 of which are reproduced from scratch), 2 sets of standard training and evaluation protocols, 15 GPU-accelerated evaluation metrics, and 3 kinds of robustness evaluation. This benchmark and codebase represent a significant leap forward in calibrating the current progress in the IMDL field and inspiring future breakthroughs. Code is available at: https://github.com/scu-zjz/IMDLBenCo.

## 1 Introduction

*"Experimentation is the ultimate arbiter of scientific truth." -* **Richard Feynman.**
The empowered image manipulation or generation models drive the Image Manipulation Detection & Localization (IMDL) task to the forefront of information forensics and security [24, 36]. While the task is occasionally referred to as "forgery detection" [15, 13] or "tamper detection"[32, 31] in literature, the consensus now favors the term IMDL [13] as the most apt descriptor for this study area. The scope of "manipulation" within IMDL bounds to partial image alterations that yield semantic discrepancies from the original content [38]. It does not pertain to purely generated images (e.g., images generated from pure text) or the application of image processing techniques that introduce noise or other non-semantical changes without altering the underlying meaning of the image [5].

---

[†]Equal contribution.
[*]Corresponding author: Jizhe Zhou (jzzhou@scu.edu.cn)

38th Conference on Neural Information Processing Systems (NeurIPS 2024) Track on Datasets and Benchmarks.

The terms "detection and localization" denote an IMDL model's dual responsibility: to conduct both image-level and pixel-level assessments. This involves a binary classification at the image level, discerning whether an input image is manipulated or authentic, and a segmentation task at the pixel level, depicting the exact manipulated areas through a mask. In short, an IMDL model shall identify semantically significant image alterations and deliver a twofold outcome: a class label and a manipulation mask.

Despite the rapid success of deep neural networks in the IMDL fields [10, 45, 11], existing models suffer from inconsistent training and evaluation protocols, supported by Tables in Appendix A.1. These inconsistencies result in incompatible and unfair comparisons, yielding insufficient and misleading experimental outcomes. Hence, establishing a unified and comprehensive benchmark is the foremost concern in the IMDL field. However, constructing this benchmark is far more than straightforward protocol unification or simply model re-training. First, the training code for most state-of-the-art (SoTA) works is not publicly available, and the source code for some SoTA works is totally unreleased [27]. Second, IMDL models commonly incorporate diverse low-level features [43, 4, 13] and complex loss functions, requiring highly customized model architecture and decoupled pipeline design for efficient reproduction. Existing frameworks, like OpenMMLab[2] and Detectron2[3], heavily rely on the registry mechanism and tightly coupled pipelines. This conflict leads to severe efficiency issues while reproducing IMDL models under existing frameworks and results in monolithic model architecture with extremely high coding load and low scalability. Consequently, a comprehensive IMDL benchmark is yet to be built.

To address this issue, we introduce IMDL-BenCo, the first comprehensive IMDL benchmark and modular codebase. IMDL-BenCo: **i)** features a modular codebase with four components: *data loader, model zoo, training script*, and *evaluator*; the model zoo contains customizable model architecture includes *SoTA models* and *backbone models*. The loss design is also isolated within the model zoo, while other components are standardized by the interface and are highly reusable; this approach mitigates conflicts between model customization and coding efficiency; **ii)** fully implements or incorporates training code for 8 SoTA IMDL models (See Table 1) and establishes a comprehensive benchmark with 2 sets of standard training and evaluation protocols, 15 GPU-accelerated evaluation metrics, and 3 kinds of robustness evaluation, and **iii)** conducts in-depth analysis based on the established benchmark and codebase, offering new insights into IMDL model architecture, dataset characteristics, and evaluation standards. This benchmark and codebase represent a significant leap forward in calibrating the current progress in the IMDL field and can inspire future breakthroughs.

## 2    Related Works

**IMDL Model Architectures.** The key to IMDL is identifying the artifacts created by manipulation. Artifacts are considered manifest on the low-level feature space. Therefore, almost all existing IMDL models share the "backbone network + low-level feature extractor" paradigm. For example, SPAN [18] and ManTra-Net [43] use VGG [34] as the backbone of their models and combine SRM [49] and BayarConv [1] filters to obtain low-level features of the image. MVSS-Net [4] combines a Sobel [35] operator, which extracts edge information, and a BayarConv on its ResNet-50 [16] backbone to extract image noise. Detailed information about each model can be found in Table 1. Various low-level feature extractors lead to various loss functions and extremely customized model architectures. As a result, reproducing IMDL models within the existing frameworks is inefficient. The high coupling between various loss functions and training architectures also makes it extremely difficult to extend different model training frameworks. The differences between training frameworks further increase the difficulty of model reproduction. This tight coupling also severely impacts algorithm innovation and rapid iteration.

**Inconsistent Training and Evaluation Protocols.** Besides model reproducing difficulties, so far, there exist multiple strikingly different protocols for training and evaluating IMDL models. MVSS-Net, CAT-Net, and TruFor were pre-trained on the ImageNet [6] dataset. SPAN [18], PSCC-Net [25], CAT-Net [22], and TruFor [22] were trained using synthetic datasets. Additionally, TruFor used a large number of original images from popular photo-sharing websites Flickr and DPReview to train its Noiseprint++ Extractor. MVSS-Net [4] and IML-ViT [27] were trained on the CASIAv2 [8]

---
[2]https://openmmlab.com/
[3]https://github.com/facebookresearch/detectron2

Table 1: Summary of the compared IMDL models

| Model Name | Venue | Backbone | Feature Extractor | Repositories | Training Code |
|---|---|---|---|---|---|
| ManTra-Net[43] | CVPR19 | VGG[34] | BayarConv+SRM Filter | https://github.com/RonyAbecidan/ManTraNet-pytorch | × |
| MVSS-Net[7] | ICCV21 | ResNet-50[16] | BayarConv+Sobel operator | https://github.com/dong03/MVSS-Net | × |
| CAT-Net[22] | IJCV22 | HRNet[39] | High-Pass Filter | https://github.com/mjkwon2021/CAT-Net | ✓ |
| ObjectFormer[40] | CVPR22 | Transformer [37] | High-Pass Filter | Unpublished code, reproduced by us | × |
| PSCC-Net[25] | TCSVT22 | HRNet[39] | Multi-Resolution Convolution Streams | https://github.com/proteus1991/PSCC-Net | ✓ |
| NCL-IML[47] | ICCV23 | ResNet-101[16] | Contrastive Learning | https://github.com/Knightzjz/NCL-IML | ✓ |
| TruFor[13] | CVPR23 | SegFormer[44] | Contrastive Learning | https://github.com/grip-unina/TruFor | × |
| IML-ViT[27] | Arxiv | Vision Transformer[9] | - | https://github.com/SunnyHaze/IML-ViT | ✓ |

dataset. on the other hand, NCL [47] did not use pre-training and was trained on the NIST16 [12] dataset. The detailed training and evaluation protocols for the models are explained in Appendix A.1. Considering the IMDL benchmark datasets are all tiny-sized (a few hundred to a few thousand images) [47], the substantial differences in training datasets make it inevitable that models using large training sets or pre-training will perform exceptionally well on other evaluation sets, posing a great challenge to the fairness of model performance evaluation. Besides, as shown in Table 1, most models do not fully open-source their code. Their results are hard to calibrate and can be highly misleading for new IMDL researchers.

**Exisiting IMDL Surveys and Benchmark.** Although IMDL surveys [28, 46] already noticed the protocol inconsistency and model reproducing difficulties in IMDL research, rare efforts have been devoted to addressing this issue. Existing surveys often rely on independently designed models with unique training strategies and datasets, leading to biases in reported results. Moreover, as far as we know, there is no comprehensive benchmark available to ensure fair and consistent evaluation of IMDL models. This absence of a unified benchmark leads to misleading, unfaithful model assessments and undermines the overall progress in the IMDL field.

# 3 Our Codebase

This section introduces our modular, research-oriented, and user-friendly codebase implemented with PyTorch[4]. As shown in Figure 1, it includes four key components: *data loader*, *model zoo*, *training script*, and *evaluator*. Our codebase strikes a balance between providing a standardized workflow for IMDL tasks and offering users extensive customization options to meet their specific needs.

## 3.1 Data Loader

The Data loader primarily handles dataset arrangement, augmentation, and transformation processes.

**Dataset Management.** We provide conversion scripts for each dataset to rearrange them into a set of *JSON* files. Subsequent training and evaluation can be carried out based on these *JSON* files.

**Augmentations and Transformations.** Due to the need for expert annotations and substantial manual effort, IMDL datasets are often very small, making it difficult to meet the demands of increasingly larger models. Therefore, data augmentation is essential. Additionally, it is crucial to ensure that the input modalities and image shapes meet existing models' requirements. Our data loader is designed with the following sequence of transformations: 1) **IMDL-specific transforms**: Inspired by MVSS-Net [4], we implemented naive inpainting and naive copy-move transforms, which can effectively enhance performance without extra datasets. 2) **Common transforms**: This includes typical visual transformations such as flipping, rotating, and random brightness adjustments, implemented using the Albumentations [3] library. 3) **Post transforms**: Some models require additional information other than RGB modality. For instance, CAT-Net [22] needs specific metadata unique to the JPEG format, which can be further obtained from the RGB domain from augmented images with callback functions. 4) **Shape transforms**: This includes zero-padding [27], cropping and resizing to ensure uniform input shapes. Additionally, the *Evaluators* can automatically adapt to different shaping strategies to complete metric calculations.

---

[4]https://pytorch.org/

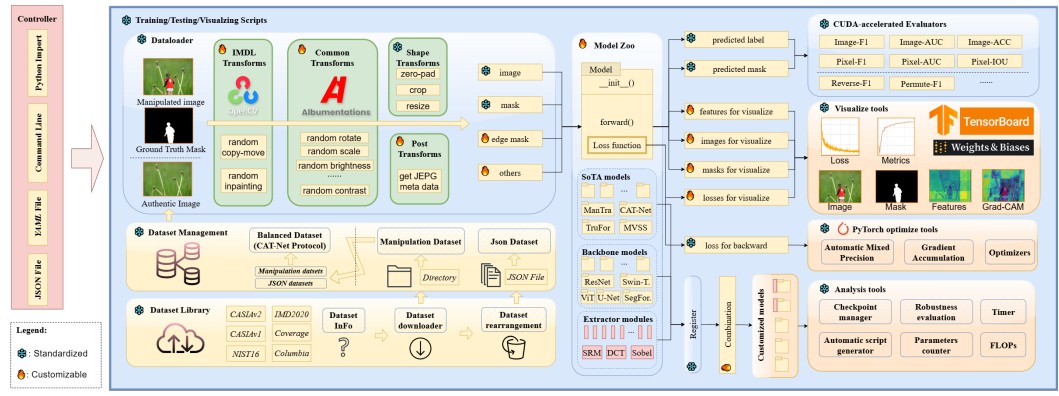

Figure 1: Overview of the paradigm for IMDL-BenCo.

## 3.2 Model Zoo

The model zoo currently consists of 8 *SoTA models*, 6 *backbone models* built with out-of-the-box vision backbones and 5 *feature extractor modules* commonly used for IMDL tasks. It is important to emphasize that we aim for all models to be trained using the same *training script* (standardization), while also being adaptable to all SoTA IMDL models (customization). As shown in Figure 1, we integrate the loss function computation within the forward function of the model. Through a unified interface, we pass in the required information, such as images and masks, and output the prediction results, the loss for backpropagation, and any losses, intermediate features, and images that need to be visualized. Therefore, for new IMDL methods, users only need to add the model scripts with loss design into the model zoo, and it will seamlessly integrate with all other components in IMDL-BenCo. By effectively reproducing current SoTA models using this framework, we demonstrate that we have successfully balanced the conflict between standardization and customization.

1) **SoTA models.** As shown in Table 1, we have faithfully reproduced 8 mainstream IMDL SoTA models, adhering to the settings from the original work. Wherever possible, we used the publicly available code, making only the necessary interface modifications. For models lacking publicly available code, we implemented them based on the settings described in their respective papers. Implementation details for each model are listed in Appendix A.3. 2) **Backbone models**: As classification and segmentation tasks, mainstream IMDL algorithms make extensive use of existing visual backbones, and the performance of these backbones also impacts the performance of the implemented algorithms. Therefore, we have adapted widely used vision backbones including ResNet [16], U-Net [30], ViT [9], Swin-Transformer [26], and SegFormer [44] into IMDL-BenCo as backbones. 3) **Feature extractor modules**: Currently, several standard feature extractors are widely used in IMDL tasks. We have implemented 5 mainstream feature extractors as $nn.module$, which include discrete cosine transform (DCT), fast Fourier transform (FFT) [2], Sobel operator [35], BayarConv [1], and SRM filter [49]—allowing seamless integration with our backbone models with registry mechanism for managing large-scale experiments or import directly for convenient use in subsequent research.

## 3.3 Training Scripts

The *training scripts* are the entry point for using IMDL-BenCo, integrating other *components* to perform specific functions. It can efficiently automate tasks such as model training, metrics evaluation, visualization, GradCAM analysing [33], and complexity computing based on configuration files (e.g., JSON, command line, or YAML). To avoid the high coupling of training pipelines seen in other frameworks (e.g., Open MM Lab often requires modifying Python package functions to customize features), we provide a code generator that allows users to create highly customized training scripts while still leveraging IMDL-BenCo's efficient components to enhance development efficiency.

## 3.4 Evaluators

Evaluation metrics are crucial for assessing the performance of IMDL models. Yet, existing methods face two key issues: 1) metrics are often unclear, with terms like optimal-F1 [4], permute-F1 [22, 13], micro-F1 and macro-F1[5] used as F1 score anonymously, and 2) most open-source codes compute metrics on the CPU, resulting in slow processing speeds.

To address these problems, we developed GPU-accelerated *evaluators* in PyTorch, integrated as standard metrics. Each *evaluator* computes a specific metric, including image-level (detection) F1 score, AUC (area-under-curve), accuracy; and pixel-level (localization) F1 score, AUC, accuracy, and IOU (Intersection over Union). All algorithms automatically adapt to *shape transformations* in the *data loader*, providing added convenience. We also explicitly implemented derived algorithms such as inverse-F1 and permute-F1 to evaluate their tendency for **overestimation**, as demonstrated in Section 5.3. This underscores the importance of precise and transparent metric selection in future work to ensure fair and consistent comparisons.

We experimented with 12,554 images from the CASIAv2 dataset and four NVIDIA 4090 GPUs and tested our evaluators' time efficiency with $nn.Identity$ as the model, which incurs negligible computation time. The results, shown in Table 2, indicate that our algorithms significantly reduce metric evaluation time, providing a faster and more reliable tool for large-scale IMDL tasks.

Table 2: Evaluator Accelerate Comparison on 12,554 images (HH:MM:SS)

| Resolution | Method | Pixel-level | | | | Image-level | | |
|---|---|---|---|---|---|---|---|---|
| | | F1 | AUC | ACC | IOU | F1 | AUC | ACC |
| 512×512 | Sklearn | 0:07:50 | 0:07:33 | 0:07:32 | 0:07:26 | 0:00:43 | 0:00:34 | 0:00:42 |
| | *IMDL-BenCo (Ours)* | 0:00:26 | 0:00:27 | 0:00:31 | 0:00:29 | 0:00:29 | 0:00:29 | 0:00:32 |
| 1024×1024 | Sklearn | 0:14:03 | 0:14:51 | 0:14:07 | 0:17:09 | 0:02:41 | 0:02:31 | 0:02:44 |
| | *IMDL-BenCo (Ours)* | 0:01:32 | 0:01:31 | 0:01:37 | 0:01:45 | 0:01:38 | 0:01:22 | 0:01:29 |

# 4 Our Benchmark

## 4.1 Benchmark Settings

**Datasets.** Our benchmark includes eight publicly available datasets frequently used in the IMDL field: CASIA[8], Fantastic Reality[21], IMD2020[29], NIST16[12], Columbia[17], COVERAGE[42], tampered COCO[22], tampered RAISE[22]. Details of each dataset are shown in Appendix A.2.

**Evaluation Metrics.** Since manipulated regions are often smaller than authentic ones, the pixel-level F1 score is widely used as a suitable metric for evaluating model performance. We assess each model's pixel-level F1 score using a fixed threshold of 0.5 across two protocols. We also evaluate all models using pixel-level AUC and IOU metrics. For models with a detection head, we additionally report image-level F1 scores. Lastly, we present robustness test results for the pixel-level F1 score under conditions of Gaussian blur, Gaussian noise, and JPEG compression.

**Hardware Configurations.** The experiments are conducted on three distinct servers with two AMD EPYC 7542 CPUs and 128G RAM, and contain 4×NVIDIA A40 GPUs, 6×NVIDIA 3090 GPUs, and 4×NVIDIA 4090 GPUs, respectively.

**Models and Hyperparameters.** Our benchmark selects eight SoTA methods in the IMDL field as the initial batch of implemented methods. The models are: Mantra-Net[43], MVSS-Net[4], CAT-Net[22], ObjectFormer[40], PSCC-Net[25], NCL-IML[47], Trufor[13], and IML-ViT[27]. The details of our minor modification, settings, and hyperparameters can be found in Appendix A.3.

## 4.2 Benchmark Protocols

The imbalance in scale and quality of existing public datasets has led to inconsistencies in the training and evaluation protocols of IMDL methods. This makes it difficult to achieve fair and convenient comparisons between existing methods. To this end, we select two reasonable and widely used protocols: **Protocol-MVSS**, proposed by MVSS-Net[4], where the model is trained only on the

---
[5]`https://scikit-learn.org/stable/modules/generated/sklearn.metrics.f1_score.html#sklearn.metrics.f1_score`

CASIAv2 dataset and then tested directly on other datasets without fine-tuning. The CASIAv2 dataset is of moderate size but high quality, making this protocol a good measure of the model's generalization ability. **Protocol-CAT**, proposed by CAT-Net[22], where the model is trained on a mixed dataset consisting of CASIAv2, Fantastic Reality, IMD2020, tampered COCO, and tampered RAISE. For each epoch, a fixed number of images are randomly sampled for training, and the model is then tested directly on other datasets without fine-tuning. This mixed dataset includes various types of tampering and a large number of manipulated images, allowing the model to demonstrate its learning capabilities effectively.

Specifically, in Protocol-MVSS, we do not use authentic images during training for methods that do not have a detection head. While in Protocol-CAT, we do not make this distinction. Additionally, we use the recommended image scaling sizes specified in each paper. We also apply consistent data augmentation techniques for all methods, such as flipping, blurring, compression and simple copy-move and inpainting operations implemented using OpenCV[6].

### 4.3 Our Benchmark Results

For Protocol-MVSS, we report the F1 scores on the COVERAGE, Columbia, NIST16, CASIAv1, and IMD2020 datasets. For Protocol-CAT, we exclude IMD2020 as it is included in the training process. The pixel-level F1 scores and image-level F1 scores are presented in Table 3 and Table 4, respectively. The pixel-level AUC and IoU can be found in Appendix A.4. There are some discrepancies between the performance of our reproduced models and the results reported in the original papers. Additionally, some models showed limited performance when compared under the same protocol. We also depict the robustness test under Protocol-MVSS in Figure 2. We observed significant robustness variations in some models compared to their original papers, demonstrating the necessity and fidelity of a unified framework.

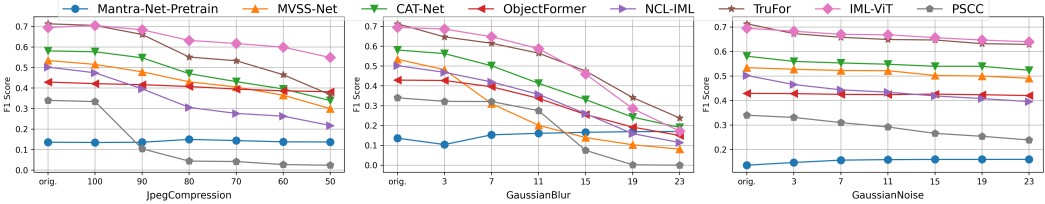

Figure 2: Results for robustness evaluation under Protocol-MVSS.

Table 3: **Pixel-level Performance.** *DH* means the model includes a detection head. When testing pixel-level F1 scores using the datasets below, we did not use authentic images. The best and second-best performances for each dataset under each protocol are highlighted in **bold** and underlined.

| Protocol | Method | DH | Size | COVERAGE | Columbia | NIST16 | CASIAv1 | IMD2020 | Average |
|----------|--------|----|------|----------|----------|--------|---------|---------|---------|
| Protocol-MVSS | Mantra-Net[43] | × | 256×256 | 0.090 | 0.243 | 0.104 | 0.125 | 0.055 | 0.123 |
| | MVSS-Net[4] | ✓ | 512×512 | 0.259 | 0.386 | 0.246 | 0.534 | 0.279 | 0.341 |
| | CAT-Net[22] | × | 512×512 | 0.296 | 0.584 | 0.269 | 0.581 | 0.273 | 0.401 |
| | ObjectFormer[40] | × | 224×224 | 0.294 | 0.336 | 0.173 | 0.429 | 0.173 | 0.281 |
| | PSCC-Net[25] | ✓ | 256×256 | 0.231 | 0.604 | 0.214 | 0.378 | 0.235 | 0.333 |
| | NCL-IML[47] | × | 512×512 | 0.225 | 0.446 | 0.260 | 0.502 | 0.237 | 0.334 |
| | Trufor[13] | ✓ | 512×512 | 0.419 | **0.865** | 0.324 | **0.721** | 0.322 | **0.530** |
| | IML-ViT[27] | × | 1024×1024 | **0.435** | 0.780 | **0.331** | **0.721** | **0.327** | 0.519 |
| Protocol-CAT | Mantra-Net[43] | × | 256×256 | 0.196 | 0.462 | 0.193 | 0.327 | × | 0.295 |
| | MVSS-Net[4] | ✓ | 512×512 | 0.498 | 0.739 | 0.348 | 0.603 | × | 0.547 |
| | CAT-Net[22] | × | 512×512 | 0.427 | 0.915 | 0.252 | 0.808 | × | 0.601 |
| | ObjectFormer[40] | × | 224×224 | 0.257 | 0.732 | 0.268 | 0.531 | × | 0.447 |
| | PSCC-Net[25] | ✓ | 256×256 | 0.379 | 0.864 | 0.369 | 0.592 | × | 0.551 |
| | NCL-IML[47] | × | 512×512 | 0.320 | 0.657 | 0.315 | 0.570 | × | 0.467 |
| | Trufor[13] | ✓ | 512×512 | 0.457 | 0.885 | 0.348 | **0.818** | × | 0.627 |
| | IML-ViT[27] | × | 1024×1024 | **0.654** | **0.948** | **0.501** | 0.795 | × | **0.725** |

---

[6]https://opencv.org/

Table 4: **Image-level Performance.** CASIAv1 is removed because it is missing authentic images.

| Protocol | Method | Size | COVERAGE | Columbia | NIST16 | IMD2020 | Average |
|---|---|---|---|---|---|---|---|
| | MVSS-Net[4] | 512×512 | 0.567 | 0.648 | 0.562 | 0.745 | 0.631 |
| Protocol-MVSS | PSCC-Net[25] | 256×256 | 0.533 | 0.747 | 0.521 | 0.684 | 0.621 |
| | Trufor[13] | 512×512 | 0.524 | 0.799 | 0.531 | 0.538 | 0.598 |
| | MVSS-Net[4] | 512×512 | 0.666 | 0.650 | 0.585 | × | 0.634 |
| Protocol-CAT | PSCC-Net[25] | 256×256 | 0.525 | 0.722 | 0.523 | × | 0.590 |
| | Trufor[13] | 512×512 | 0.645 | 0.934 | 0.638 | × | 0.739 |

Table 5: **Backbone parameters and floating-point operations per second (FLOPs).** In ViT-B/16-8-cat, "8" refers to the first eight layers of transformer blocks, while "cat" denotes the concatenation of the feature and original image feature along the sequence dimension before input into the blocks.

| Backbone | ResNet151[16] | U-Net[30] | ViT-B/16[9] | Swin-B[26] | ViT-B/16-8[9] | ViT-B/16-8-cat[9] | SegFormer-B2[44] |
|---|---|---|---|---|---|---|---|
| Parameters | 48.030M | 31.038M | 89.343M | 90.515M | 60.991M | 62.368M | 25.764M |
| FLOPs | 65.144G | 197.632G | 92.026G | 88.565G | 62.975G | 124.928G | 21.562G |

# 5 Experiments and Analysis

Our codebase and benchmark unify testing and training protocols for IMDL models. Despite this unification, the IMDL task also retains multiple unique and critical characteristics compared to other detection or segmentation tasks, notably the reliance on the "backbone network + low-level feature extractor" paradigm, benchmark datasets with random splits, and various evaluation metrics. Accordingly, we further investigate and deeply analyze 4 widely concerned but less-explored questions in sequence, including: **1)** *Is the low-level feature extractor a must in IMDL?* **2)** *Which backbone architecture best fits the IMDL task?* **3)** *Do random split training and testing datasets affect the model performances?* **4)** *What metrics mostly represent the model's practical behavior?*

Through extensive experiments, we are the first to answer the above question with evidential facts and provide new critical insights into model design, dataset cleansing, and metrics for the IMDL field.

## 5.1 Low-level Feature Extractors and Backbones

As shown in Table 1, prevailing IMDL approaches heavily rely on feature extractors to detect manipulation traces. However, few articles specifically analyze the advantages of different extractors. In this section, we combine the *backbone models* implemented in the *model zoo* (see Section 3.2) with different *feature extractor modules* to explore the performance of each feature extractor and their compatibility with the backbone. The complexity of each combined model is shown in Table 5.

All combined models are trained on the CASIAv2 dataset for 200 epochs with an image size of 512×512. Detailed experimental settings can be found in Appendix A.7.1. They are then evaluated on four distinct datasets—CASIAv1, Columbia, NIST16, Coverage, and IMD2020—to assess their generalization capabilities. The average F1 score for each setting across the four datasets is reported in Table 6.

Experiments indicate that specific feature extractors, such as BayarConv and Sobel, can negatively impact model performance. In contrast, the ResNet model, when equipped with DCT, FFT, and SRM feature extractors, shows improved performance. The ViT model, when equipped with feature extractors, tends to underfit and may require more epochs to converge. A better feature fusion

Table 6: **Comparison of Generalization Performance** Across Different Backbones and Feature Extractors. The Average F1 Score represents the mean F1 score across five datasets.

| Backbone | ResNet151[16] | | | | | | U-Net[30] | | | | | |
|---|---|---|---|---|---|---|---|---|---|---|---|---|
| Feature Extractor | None | Bayar | DCT | FFT | Sobel | SRM | None | Bayar. | DCT | FFT | Sobel | SRM |
| Average F1 Score | 0.354 | 0.227 | 0.343 | 0.358 | 0.207 | 0.355 | 0.015 | 0.018 | 0.013 | 0.013 | 0.015 | 0.016 |

| Backbone | SegFormer-B2[44] | | | | | | Swin-B[26] | | | | | |
|---|---|---|---|---|---|---|---|---|---|---|---|---|
| Feature Extractor | None | Bayar | DCT | FFT | Sobel | SRM | None | Bayar. | DCT | FFT | Sobel | SRM |
| Average F1 Score | 0.466 | 0.143 | 0.364 | 0.364 | 0.147 | 0.363 | 0.538 | 0.214 | 0.472 | 0.447 | 0.225 | 0.395 |

| Backbone | ViT-B/16[9] | | | | | | ViT-B/16-8[9] | | | | | |
|---|---|---|---|---|---|---|---|---|---|---|---|---|
| Feature Extractor | None | Bayar | DCT | FFT | Sobel | SRM | None | Bayar. | DCT | FFT | Sobel | SRM |
| Average F1 Score | 0.295 | 0.068 | 0.228 | 0.241 | 0.042 | 0.254 | 0.213 | 0.133 | 0.151 | 0.188 | 0.128 | 0.216 |

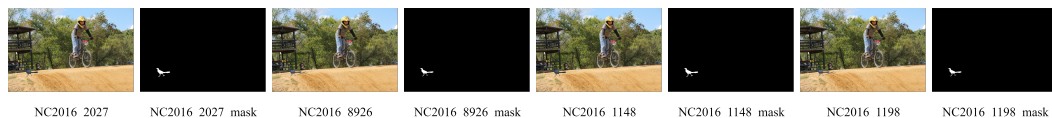

NC2016_2027  NC2016_2027_mask  NC2016_8926  NC2016_8926_mask  NC2016_1148  NC2016_1148_mask  NC2016_1198  NC2016_1198_mask

Figure 3: **An instance of label leakage in NIST16.** There is almost no visible difference between these images and their masks. When split into training and testing datasets, the test images are easily located by the model, indicating an overestimation of model performance.

Table 7: **Pixel-level performance of the Models on NIST16 and NIST16-C.** The pixel-level F1 scores for all models on the NIST16-C dataset stay mostly the same as the original NIST16 dataset.

| Protocol | Dataset | ManTra-Net[43] | MVSS-Net[7] | CAT-Net[22] | ObjectFormer[40] | PSCC-Net[25] | NCL-IML[47] | Trufor[13] | IML-ViT[27] |
|---|---|---|---|---|---|---|---|---|---|
| Protocol-MVSS | NIST16 | 0.104 | 0.2461 | 0.2692 | 0.1732 | 0.2141 | 0.2599 | 0.3241 | 0.331 |
| | NIST16-C | 0.06493 | 0.2404 | 0.2734 | 0.1922 | 0.217 | 0.2469 | 0.291 | 0.2657 |
| Protocol-CAT | NIST16 | 0.1837 | 0.3477 | 0.2522 | 0.2682 | 0.3689 | 0.3148 | 0.348 | 0.5013 |
| | NIST16-C | 0.1629 | 0.333 | 0.3318 | 0.2637 | 0.3476 | 0.301 | 0.344 | 0.4404 |

method might eliminate the current issues with the ViT model. For the Swin Transformer, adding feature extractors can lead to overfitting, while the performance of the Segformer generally degrades. Detailed discussion and experiment results are detailed in Appendix A.7.2.

**Necessity for Feature Extractors.** In short, BayarConv and Sobel are not suitable for IMDL tasks. Appropriate low-level feature extractors, such as DCT, FFT, and SRM, can enhance the performance of the ResNet model. However, all feature extractors may impede convergence for ViT and its variants, cause overfitting in Swin Transformer, and lead to an overall performance decline in SegFormer. Therefore, low-level feature extractors are not necessities in IMDL.

**Backbone Fitness.** As shown in Table 6, Swin Transformer and Segformer demonstrate robust performance on the IMDL task, outperforming ResNet and ViT. The U-Net architecture is not well-suited for this task.

## 5.2 Dataset Bias and Cleansing Methods

**Dataset Bias.** Through our benchmark, we find that comparing the model performance under our protocol with the model performance after fine-tuning in their paper, on every model, a significant performance decline is observed on the NIST16 dataset. However, such a huge decline does not occur on other benchmark datasets. After thorough analysis, we find the NIST16 dataset contains "extremely similar" manipulation patterns. Then, when these extremely similar images are randomly split into the training and testing sets, models can effectively locate the manipulation regions by memorizing the extremely similar training samples. We refer to this critical dataset bias as "label leakage." Figure 3 illustrates an instance of label leakage in the NIST16 dataset.

**Dataset Cleansing.** To enhance the reliability of NIST16 for evaluation, we introduce a new dataset, NIST16-C[7], created by applying a filtering method based on Structural Similarity (SSIM) [41]. This process helps eliminate overly similar images, reduce dataset bias, and prevent performance overestimation caused by label leakage. Further details for our analysis and the cleansing procedure can be found in Appendix A.7.3.

We conducted extensive benchmarking on NIST16-C, and the test results are shown in Table 7. This demonstrates that we have eliminated redundant data in the NIST16 dataset, addressing label leakage. As a result, models can now focus on learning the underlying features of manipulations.

## 5.3 Evaluation Metrics Selection

**Controversial F1 Scores.** The F1 metric has multiple variations with different computation equations, such as invert-F1 and permute-F1[20, 22, 13]. Invert-F1 is the value obtained by calculating F1 between the inverted prediction results and the original mask. Permute-F1 is the maximum value between original F1 and invert-F1. The formula is Permute-F1$(G, P) =$

---

[7]NIST16-C is available here: https://github.com/DSLJDI/NIST16-data-set-deduplication

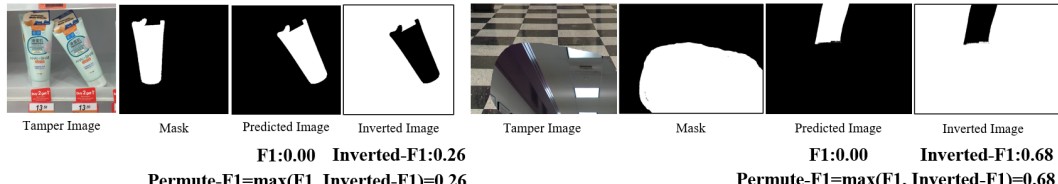



| Tamper Image | Mask | Predicted Image | Inverted Image | Tamper Image | Mask | Predicted Image | Inverted Image |

**F1:0.00    Inverted-F1:0.26**       **F1:0.00    Inverted-F1:0.68**

**Permute-F1=max(F1, Inverted-F1)=0.26**   **Permute-F1=max(F1, Inverted-F1)=0.68**



Figure 4: Controversial permuted metrics. When the model's predictions are completely wrong, the F1 score should theoretically be 0.00.

$\max(\text{F1}(G, P), \text{Invert-F1}(G, P))$, where $G$ is the ground truth and $P$ is the predicted mask. As shown in Figure 4, when the white area of the mask is large, and there is a significant deviation between the model's prediction and the mask, the invert-F1 score is much higher than the F1 score. This metric affects the fairness of the evaluation.

Furthermore, using parameters such as "macro", "micro", and "weighted" when calculating F1 scores via the sklearn library is inappropriate, as it artificially inflates our F1 metrics, which is unjustified. We further analyze these misleading F1 metrics in Appendix A.7.4. Mixing these F1 scores anonymously would lead to significant fairness issues. In summary, we contend that using the F1 score with the "binary" parameter for calculation is more scientific and rigorous. We hope that future research will uniformly adopt this standard. Additionally, we discuss the current issue of AUC being overestimated in the Appendix A.7.5.

# 6  Conclusions

In conclusion, IMDL-BenCo marks a significant advancement in the field of Image Manipulation Detection & Localization. By offering a comprehensive benchmark and a modular codebase, IMDL-BenCo enhances coding efficiency and customization flexibility, thus facilitating rigorous experimentation and fair comparison of IMDL models. Besides, IMDL-BenCo inspires future breakthroughs by providing a unified and scalable framework for model development and evaluation. We anticipate that IMDL-BenCo will become an essential resource for researchers and practitioners, driving forward the capabilities and applications of IMDL technologies in various fields, including information forensics and security.

# 7  Author Contributions

The authors' contributions are: **Xiaochen Ma**: codebase design, the coding leader, and manuscript writing. **Xuekang Zhu**: implements *ObjectFormer*, *backbone models*, and *extractor modules*; and manuscript writing. **Lei Su**: implements *MVSS-Net*, *NCL-IML*, *ManTra-Net*, *and cleaned NIST16*; and manuscript writing. **Bo Du**: implements *PSCC-Net*, *Trufor*, and manuscript writing. **Zhuohang Jiang**: implements GPU-accelerated *evaluators*, and manuscript writing. **Bingkui Tong**: implements *CAT-Net*, Grad-CAM, and manuscript writing. **Zeyu Lei**: implements *ManTra-Net*, *SPAN*, and manuscript writing. **Xinyu Yang**: dataset debiasing, metrics analyzing, and manuscript writing. **Jiancheng Lv**: general project advising. **Chi-Man Pun**: project advising. **Jizhe Zhou**: project supervisor and manuscript writing.

# 8  Acknowledgement

This research was supported by the Sichuan Provincial Natural Science Foundation (Grant No.2024YFHZ0355), the Fundamental Research Funds for the Central Universities (Grant No.2022SCU12072 and No.YJ2021159), and the Science and Technology Development Fund, Macau SAR (Grant 0141/2023/RIA2 and 0193/2023/RIA3). The authors would like to give special thanks to *Kaiwen Feng* for his attentive work in analyzing the macro-F1 issues and fixing bugs on the IMDL-BenCo codebase and *Dr. Wentao Feng* for the workplace, computation power, and physical infrastructure support.

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

# A   Appendix

## A.1   Differences in Pretraining, Training, Evaluation, Metrics between SoTA models

As shown in Table 8, existing IMDL models exhibit significant discrepancies between the pre-training datasets used (emphasizing the pre-training of the backbone, typically on natural images like ImageNet [6] or MS-COCO [23]) and the specific datasets used for training in the IMDL tasks. This introduces unfairness. Furthermore, as indicated in Table 9, there is a considerable gap between the datasets selected for testing and the evaluation metrics used during testing. Therefore, to ensure the healthy and sustainable development of this field, it is crucial to establish a standardized benchmark for the fair and scientific evaluation of IMDL models' performance.

| Model Name | Pretraining Dataset | | | | Training Dataset | | | | | | | | | | | | |
|---|---|---|---|---|---|---|---|---|---|---|---|---|---|---|---|---|---|
| | IN | SD1 | TF | CO | DID | KCMI | SD2 | CA2 | DT | FR | Cov | IMD | CO+ | RA+ | NI16 | Pa+ | Col |
| ManTra-Net | ✓ | - | - | - | ✓ | ✓ | ✓ | - | - | - | - | - | - | - | - | - | - |
| SPAN | ✓ | - | - | - | - | - | ✓ | - | - | - | - | - | - | - | - | - | - |
| MVSS-Net | ✓ | - | - | - | - | - | - | ✓ | ✓ | - | - | - | - | - | - | - | - |
| CAT-Net | ✓ | - | - | - | - | - | - | ✓ | - | ✓ | - | ✓ | ✓ | ✓ | - | - | - |
| ObjectFormer | - | ✓ | - | - | - | - | - | - | - | - | - | - | ✓ | - | - | ✓ | - |
| TruFor | ✓ | - | ✓ | - | - | - | - | ✓ | - | ✓ | - | ✓ | ✓ | ✓ | - | - | - |
| PSCC-Net | - | - | - | ✓ | - | - | - | - | - | - | - | - | ✓ | - | - | - | - |
| IML-ViT | ✓ | - | - | - | - | - | - | ✓ | - | - | - | - | - | - | - | - | - |
| PMAE | ✓ | - | - | - | - | - | - | ✓ | - | - | - | - | - | - | - | - | - |
| NCL-IML | - | - | - | - | - | - | - | ✓ | ✓ | - | ✓ | - | - | - | ✓ | - | ✓ |
| | | | | | | | | | | | | | | | | | - |

Table 8: The information about the pretraining and training datasets for the models we reproduced. IN, SD1, TF, CO, DID, KCMI, SD2, CA2, DT, FR, Cov, IMD, CO+, RA+, NI16, Pa+, and Col correspond to ImageNet, Synthetic Dataset 1, Trufor dataset for noiseprint++, COCO dataset, Dresden Image Database, Kaggle Camera Model Identification, Synthetic Dataset 2, CASIAv2, DEFACTO, FantasticReality, Coverage, IMD2020, COCO+ (a tampered dataset generated from COCO), RAISE+ (a tampered dataset generated from RAISE), NIST16, Paris+ (a tampered dataset generated from Paris), and Columbia, respectively.

| Model Name | Evaluating Dataset | | | | | | | | | | | | | | Evaluating Metrics | | | | | | | |
|---|---|---|---|---|---|---|---|---|---|---|---|---|---|---|---|---|---|---|---|---|---|---|
| | NI16 | Col | CA | PSB | COV | DT | Car | GRIP | CMF | IMD | DSO | VP | OF | CoG | AUC | F1 | Acc | AP | TNR | TPR | EER | TPR1% |
| ManTra-Net | ✓ | ✓ | ✓ | ✓ | ✓ | - | - | - | - | - | - | - | - | - | ✓ | ✓ | - | - | - | - | - | - |
| SPAN | ✓ | ✓ | ✓ | - | ✓ | - | - | - | - | - | - | - | - | - | ✓ | ✓ | - | - | - | - | - | - |
| MVSS-Net | ✓ | ✓ | ✓ | - | ✓ | ✓ | - | - | - | - | - | - | - | - | ✓ | ✓ | - | - | - | - | - | - |
| CAT-Net | ✓ | ✓ | ✓ | - | - | - | ✓ | ✓ | ✓ | - | - | - | - | - | - | ✓ | ✓ | ✓ | - | - | - | - |
| ObjectFormer | ✓ | ✓ | ✓ | - | - | - | - | - | ✓ | - | - | - | - | - | ✓ | ✓ | - | - | - | - | - | - |
| TruFor | ✓ | ✓ | ✓ | - | ✓ | - | - | - | - | - | ✓ | ✓ | ✓ | ✓ | ✓ | ✓ | - | - | ✓ | ✓ | - | - |
| PSCC-Net | ✓ | ✓ | ✓ | - | ✓ | - | - | - | - | ✓ | - | - | - | - | ✓ | ✓ | - | - | - | - | ✓ | ✓ |
| IML-ViT | ✓ | ✓ | ✓ | - | ✓ | ✓ | - | - | - | - | - | - | - | - | ✓ | ✓ | - | - | - | - | - | - |
| PMAE | ✓ | ✓ | ✓ | - | ✓ | - | - | - | - | - | - | - | - | - | ✓ | ✓ | - | - | - | - | - | - |
| NCL-IML | ✓ | ✓ | ✓ | - | ✓ | ✓ | - | - | - | - | - | - | - | - | ✓ | ✓ | - | - | - | - | - | - |

Table 9: Here is the validation dataset test metric information for the models we have reproduced. NI16, Col, CA, PSB, COV, DT, Car, GRIP, CMF, IMD, DSO, VP, OF, CoG respectively represent NIST16, Columbia, CASIA, PhotoShop-battle, COVERAGE, DEFACTO, Carvalho, GRIP Dataset, CoMoFoD, IMD2020, DSO-1, VIPP, OpenForensics, and CocoGlide.

## A.2   Datasets in this paper

We follow MVSS-Protocol and CAT-Protocol to evaluate the performance of all models. Thus, eight publicly available datasets are included in this benchmark. The manipulation type, the number of manipulated images, and the number of authentic images are shown in Table 10.

## A.3   SoTA models in model zoo

**SPAN** [18] SPAN uses the pre-trained VGG-based feature extractor in ManTra-Net[43] and proposes pyramid spatial attention propagation. Pyramid spatial attention propagation goes through self-attention blocks with proper dilation distances. Thus, information from each pixel location is propagated in a pyramid structure. Then, it uses three convolutional layers to make the decision. We use the feature extractor and its weight from ManTraNet-pytorch[8], then implement the PyTorch

---

[8] https://github.com/RonyAbecidan/ManTraNet-pytorch

Table 10: **Datasets information.** The composition of images and types of manipulation images in various datasets."×" means there is no relevant information.

| Dataset | Type | | Manipulation type | | |
|---|---|---|---|---|---|
| | auth | tamp | cope-move | splice | remove |
| CASIA v2 | 7491 | 5123 | 3274 | 1849 | 0 |
| CASIA v1 | 0 | 920 | 459 | 461 | 0 |
| IMD2020 | 414 | 2010 | × | × | × |
| NIST16 | 0 | 564 | 236 | 225 | 103 |
| COVERAGE | 100 | 100 | 100 | 0 | 0 |
| Columbia | 183 | 180 | 0 | 180 | 0 |
| Fantastic Reality | 16592 | 19423 | × | × | × |
| tampCOCO | 0 | 800000 | × | × | × |
| compRAISE | 24462 | 0 | 0 | 0 | 0 |

version of the pyramid spatial attention propagation and decision module. Besides, we add residual net to the VGG backbone in the feature extractor, as we find the model hard to converge.

**ManTra-Net** [43] ManTra-Net-pretrain aligns ManTraNet-pytorch[8] with our benchmark. We denormalize the input images and transfer the weight into our form (Since the normalization in the original repository differs), then test on five datasets.

**MVSS-Net** [4] MVSS-Net uses BayarConv and SRM operator as feature extractors, with ResNet50 as its backbone network. During replication, we used the code from the repository mentioned in the main text, making no major modifications. We did not use AMP (Automatic Mixed Precision) during training since it may cause NAN issues. The model was trained for 200 epochs with an initial learning rate of 1e-4. MVSS-Net performs label prediction.

**CAT-Net** [22] The CAT-Net model first extracts features through convolution from both the RGB Stream and the DCT Stream, then fuses the information from these two parts. During our replication process, we primarily used the official implementation of the model structure. However, we reimplement the artifact learning module in PyTorch to leverage GPU acceleration. This model does not include a label prediction function. The total number of training epochs was set to 200, with an initial learning rate of 0.0005.

**NCL-IML** [47] NCL-IML does not use a feature extractor and employs patch-level contrastive supervision learning, with ResNet101 as its backbone network. We used the code from the repository mentioned in the main text for replication. For loss calculation, we replaced the ASPP output loss with edge loss. We did not use AMP during training. The model was trained for 500 epochs using the SGD optimizer, with an initial learning rate of 7e-3.

**PSCC-Net** [25] PSCC-Net leverages features at different scales with dense cross-connections to produce manipulation masks in a coarse-to-fine fashion, with a lightweight backbone named HRNet. We use the model and training code from the official repository. The training is conducted for 150 epochs, with the initial learning rate reduced from 1e-4 to 0. Following the original paper, the loss function consisted of two parts: mask loss and label loss.

**Trufor** [13] Trufor relies on the extraction of both high-level and low-level traces through a transformer-based fusion architecture that combines the RGB image and a learned noise-sensitive fingerprint. The learning process consists of three stages: noiseprint++, localization, and detection. Due to the lack of pretraining data for noiseprint++, we carefully extract the weights of noiseprint++ from the checkpoint provided in the official repository to train the latter two stages. During the localization training stage, we use a learning rate of 4e-5, decaying to 0, and train for 150 epochs. In the detection training stage, we use a learning rate of 1e-4, decaying to 0, and train for 100 epochs.

**IML-ViT** [13] IML-ViT is a plain ViT structure that employs a high-resolution image input strategy supplemented by multi-scale and edge supervision. This allows for the effective utilization of various self-supervised pre-trained ViTs for initialization. It does not use any existing feature extractors. We directly used the author's original GitHub repository for the localization task, and it converged after training for 200 epochs with a learning rate of 1e-4.

**Objectformer** [40] ObjectFormer uses the transformer block from the pre-trained ViT-B/16 model as the backbone. The input image size is 224. Data preprocessing involves using the FFT package to extract high-frequency information. The image and high-frequency data are passed to their respective patch embed layers and concatenated along the sequence length dimension. ObjectFormer initializes the first eight layers of the same pre-trained transformer blocks twice for the encoder and decoder parts. In the encoder, a trainable parameter serves as the query, with the concatenated features as the key and value. The encoder's output functions as the query, key, and value for the decoder. The feature then undergoes a BCIM feature change, completing the encoder and decoder forward propagation process. After repeating this process eight times, deconvolution and linear interpolation are applied, and the output is reshaped into the mask required for prediction during training. Since the model released in the official repository differs from the structure mentioned in the original paper, we manually re-implement the ObjectFormer following the description from the paper. ObjectFormer uses a cosine learning rate schedule starting from 1e-5 to 5-7, training the model for 1100 epochs on the Protocol-MVSS dataset and 2000 epochs on the Protocol-CAT dataset.

### A.4 More Pixel-level Performance Results

In addition to the F1 score, we also evaluated the SoTA models on pixel-level AUC and IoU, with the results presented in Table 11 and Table 12, respectively. As mentioned earlier, AUC often reflects an overoptimistic metric and struggles to accurately measure the localization precision of the model. On the other hand, IoU fails to handle the extreme imbalance between negative and positive samples, which is a common issue in IMDL datasets, where the tampered area in an image is typically quite small. Therefore, we believe that the F1 score better reflects the model's localization performance.

Table 11: **Pixel-level AUC.**

| Protocol | Method | DH | Size | COVERAGE | Columbia | NIST16 | CASIAv1 | IMD2020 | Average |
|---|---|---|---|---|---|---|---|---|---|
| Protocol-MVSS | Mantra-Net[43] | × | 256×256 | 0.778 | 0.804 | 0.769 | 0.788 | 0.775 | 0.783 |
| | MVSS-Net[4] | ✓ | 512×512 | 0.720 | 0.732 | 0.737 | 0.861 | 0.755 | 0.761 |
| | CAT-Net[22] | × | 512×512 | 0.759 | 0.800 | 0.787 | 0.910 | 0.775 | 0.806 |
| | ObjectFormer[40] | × | 224×224 | 0.739 | 0.528 | 0.722 | 0.876 | 0.680 | 0.709 |
| | PSCC-Net[25] | ✓ | 256×256 | 0.697 | 0.814 | 0.725 | 0.833 | 0.778 | 0.769 |
| | Trufor[13] | ✓ | 512×512 | 0.911 | 0.928 | 0.820 | 0.946 | 0.840 | 0.889 |
| | IML-ViT[27] | × | 1024×1024 | 0.871 | 0.898 | 0.789 | 0.940 | 0.814 | 0.862 |
| Protocol-CAT | MVSS-Net[4] | ✓ | 512×512 | 0.870 | 0.933 | 0.790 | 0.912 | × | 0.876 |
| | CAT-Net[22] | × | 512×512 | 0.917 | 0.946 | 0.822 | 0.980 | × | 0.916 |
| | ObjectFormer[40] | × | 224×224 | 0.747 | 0.858 | 0.775 | 0.884 | × | 0.816 |
| | PSCC-Net[25] | ✓ | 256×256 | 0.884 | 0.946 | 0.828 | 0.919 | × | 0.894 |
| | Trufor[13] | ✓ | 512×512 | 0.942 | 0.899 | 0.878 | 0.974 | × | 0.924 |
| | IML-ViT[27] | × | 1024×1024 | 0.942 | 0.955 | 0.893 | 0.976 | × | 0.942 |

Table 12: **Pixel-level IoU.**

| Protocol | Method | DH | Size | COVERAGE | Columbia | NIST16 | CASIAv1 | IMD2020 | Average |
|---|---|---|---|---|---|---|---|---|---|
| Protocol-MVSS | Mantra-Net[43] | × | 256×256 | 0.066 | 0.348 | 0.135 | 0.092 | 0.062 | 0.141 |
| | MVSS-Net[4] | ✓ | 512×512 | 0.192 | 0.290 | 0.176 | 0.440 | 0.201 | 0.260 |
| | CAT-Net[22] | × | 512×512 | 0.204 | 0.469 | 0.196 | 0.509 | 0.198 | 0.315 |
| | ObjectFormer[40] | × | 224×224 | 0.181 | 0.224 | 0.111 | 0.320 | 0.106 | 0.478 |
| | PSCC-Net[25] | ✓ | 256×256 | 0.146 | 0.477 | 0.150 | 0.310 | 0.157 | 0.248 |
| | Trufor[13] | ✓ | 512×512 | 0.352 | 0.832 | 0.271 | 0.666 | 0.267 | 0.478 |
| | IML-ViT[27] | × | 1024×1024 | 0.372 | 0.685 | 0.250 | 0.648 | 0.264 | 0.444 |
| Protocol-CAT | MVSS-Net[4] | ✓ | 512×512 | 0.389 | 0.672 | 0.259 | 0.491 | × | 0.453 |
| | CAT-Net[22] | × | 512×512 | 0.387 | 0.895 | 0.212 | 0.748 | × | 0.561 |
| | ObjectFormer[40] | × | 224×224 | 0.188 | 0.670 | 0.203 | 0.320 | × | 0.345 |
| | PSCC-Net[25] | ✓ | 256×256 | 0.301 | 0.814 | 0.297 | 0.500 | × | 0.478 |
| | Trufor[13] | ✓ | 512×512 | 0.415 | 0.859 | 0.296 | 0.775 | × | 0.586 |
| | IML-ViT[27] | × | 1024×1024 | 0.590 | 0.933 | 0.440 | 0.749 | × | 0.678 |

### A.5 Detailed Introduction of Evaluators

In the field of Image Manipulation Detection & Localization (IMDL), the standardization of evaluation metrics is essential for ensuring the comparability and reproducibility of research outcomes. The domain currently defines seven core evaluation metrics, categorized into two major classes: image-level and pixel-level. At the image level, the evaluation metrics include the Area Under the Curve (AUC) score, F1 score, and accuracy. The pixel-level evaluation metrics further expand to include AUC, F1 score, accuracy, and Intersection over Union (IoU).

However, due to the variability in dataset preprocessing methods in previous research, there is a certain degree of disorder in the existing evaluation system. For instance, differences in image size

adjustment (resize) and padding strategies among models can lead to discardable content in the output prediction mask. Additionally, the inconsistency in how manipulated areas are annotated across different models—some marking manipulated areas as 1 while others as 0 has resulted in inconsistencies in the calculation methods and results of the evaluation metrics, and even wholly opposite outcomes in some cases. This inconsistency greatly complicates the establishment of a unified evaluation standard and benchmark in the IMDL field.

To address this issue, we conducted in-depth research and analysis, comprehensively considering the evaluation methods of existing models, and proposed a set of unified evaluation metric calculation formulas. Our goal is to eliminate the differences between various models and datasets, ensuring the consistency and comparability of evaluation results. Furthermore, to enhance the efficiency and scalability of the evaluation process, we developed an evaluation framework based on GPU multi-card acceleration, which significantly improves the calculation speed of evaluation metrics on large-scale datasets. This framework provides researchers in the IMDL field with an efficient and unified evaluation tool. Each evaluator has *batch_update* and *epoch_update* methods, which respectively compute metrics for each batch and after each epoch, catering to different needs.

Next, we will provide a detailed introduction to our *Evaluators* and the details of their implementation.

### A.5.1 Confusion Matrix Acceleration

The confusion matrix is a critical step in computing model metrics. In sklearn, it is often computed using the CPU, which processes each image individually, resulting in slow computation speed. We found that the way to compute the confusion matrix is fixed, so it is possible to calculate multiple images simultaneously through matrix computation. The following algorithm1 uses GPU acceleration to speed up the computation of the confusion matrix.

---
**Algorithm 1** Confusion Matrix

---
1: **procedure** CAL CONFUSION MATRIX($predict, mask, shape\_mask$)
2:     **if** $shape\_mask = None$ **then**
3:         $TP = \text{sum}((1 - predict) \times mask \times shape_mask)$
4:         $TN = \text{sum}(predict \times (1 - mask) \times shape_mask)$
5:         $FP = \text{sum}((1 - predict) \times (1 - mask) \times shape_mask)$
6:         $FN = \text{sum}(predict \times mask \times shape_mask)$
7:         **return** $TP, TN, FP, FN$
8:     **else**
9:         **if** $shape\_mask! = None$ **then**
10:             $TP = \text{sum}((1 - predict) \times mask )$
11:             $TN = \text{sum}(predict \times (1 - mask) )$
12:             $FP = \text{sum}((1 - predict) \times (1 - mask))$
13:             $FN = \text{sum}(predict \times mask )$
14:             **return** $TP, TN, FP, FN$
15:         **end if**
16:     **end if**
17: **end procedure**

---

### A.5.2 Pixel-Level Evaluators Acceleration

Pixel-level evaluation includes F1, AUC, ACC, and IOU. Each of these metrics can be calculated with the help of a confusion matrix. Therefore, it is possible to compute pixel-level metrics for multiple images at the same time through matrix computation, thus speeding up the process. During the computation, we accelerate each batch and use reduction techniques externally to aggregate values computed by multiple GPUs, thereby achieving multi-GPU acceleration for Pixel-Level Evaluation.

### A.5.3 Image-Level Evaluators Acceleration

Pixel-level evaluation requires collecting predictions for all images before separately calculating F1, AUC, and ACC. Therefore, during the batch_update phase, we obtain the model's predictions on

different GPUs. In the final epoch_update phase, we use reduction and gather techniques to aggregate predictions stored on each GPU and then compute the corresponding Pixel-Level metrics.

## A.6 GradCAM for Analysis

For IMDL tasks, it is crucial to analyze whether the model captures low-level trace information or high-level semantic information to understand its decision-making process. Therefore, we used Grad-CAM [33] to examine what different models focus on in Figure 5. We found that various models prioritize different types of information in their decision support. In future research, effectively analyzing each model's performance and corresponding Grad-CAM patterns for each layer can significantly aid in designing and improving model performance.

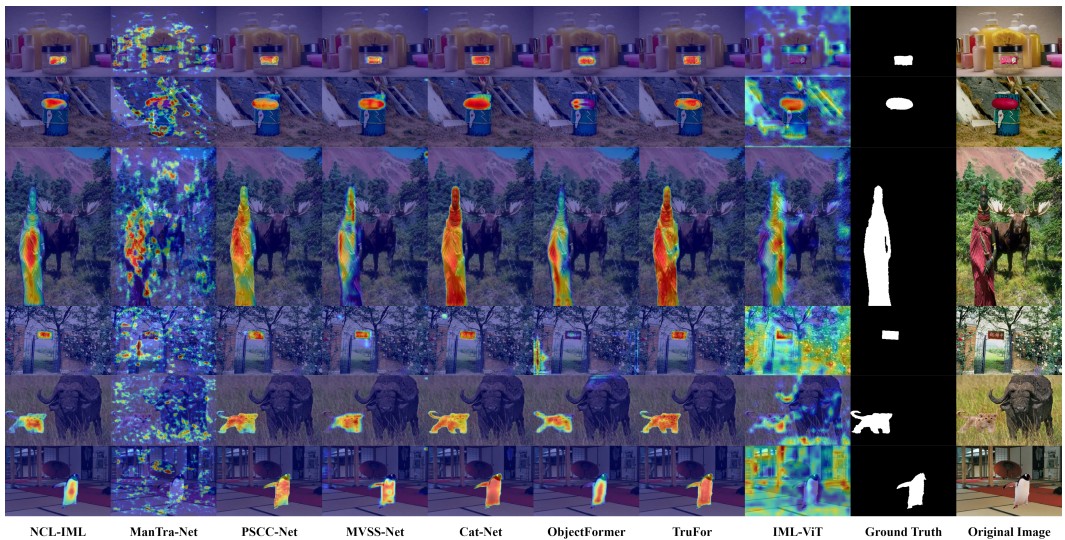

Figure 5: Grad-CAM visualization of models in our benchmark

## A.7 Additional Analysis and Insights

### A.7.1 Modules Experiments Setting

Our experiments utilized the CASIAv2 dataset for training purposes, while evaluation was conducted on four different datasets to assess the generalization capabilities: CASIAv1, Columbia, NIST16, Coverage, and IMD2020. In addition to ViT-B/16, our experiments modified the input layer to include the feature maps extracted by the feature extractors. For instance, the DCT feature extractor's output was three channels, which were concatenated with the original image to create a 6-channel input for the input layer. If the backbone output feature shape is inconsistent with the mask size, deconvolution was employed to ensure that the dimensions of the predicted mask matched the expected output. All models were initialized with pre-trained weights.

ViT-B/16-8-cat employed a distinct patch embedding technique. The original pre-trained patch embedding layer was used for patch extraction on the original image, while a new patch embedding layer was introduced for patch extraction on the feature maps generated by the feature extractors. After both layers completed patch extraction, the results were concatenated along the sequence length (L) dimension. Moreover, the combined models, based on ViT-B/16-8 and ViT-B/16-8-cat, only the first eight layers of the original transformer block were utilized to reduce training time.

In our experimental setup, all images were resized to a uniform resolution of 512×512 pixels before being fed into the models. We implemented a cosine annealing learning rate schedule, which decayed from $1e-4$ to $5e-7$ over the training period. Each experiment used a batch size of 24 and was optimized using the AdamW optimizer with weight decay set to 0.05. Additionally, we used an accumulation iteration of 8, a seed of 42, and a warmup period of 2 epochs.

Table 13: Various backbone models combined with different feature extractors. The models are trained on CASIAv2 and tested on four datasets.

| Backbone | ResNet151 | | | | | | U-Net | | | | | |
|---|---|---|---|---|---|---|---|---|---|---|---|---|
| Feature Extractor | None | Bayar | DCT | FFT | Sobel | SRM | None | Bayar | DCT | FFT | Sobel | SRM |
| CASIA v1 | 0.6286 | 0.441 | 0.6344 | 0.6337 | 0.3201 | 0.611 | 0.0227 | 0.0142 | 0.0213 | 0.0223 | 0.0152 | 0.0194 |
| Columbia | 0.4971 | 0.2274 | 0.479 | 0.5018 | 0.2487 | 0.5036 | 0.0059 | 0.0046 | 0.0048 | 0.0051 | 0.0023 | 0.0043 |
| NIST16 | 0.1864 | 0.1806 | 0.2047 | 0.2316 | 0.1994 | 0.2083 | 0.0253 | 0.0362 | 0.0215 | 0.0249 | 0.0262 | 0.0315 |
| Coverage | 0.2612 | 0.1509 | 0.2056 | 0.2323 | 0.1412 | 0.2849 | 0.0046 | 0.0273 | 0.0056 | 0.0048 | 0.0135 | 0.0108 |
| IMD20 | 0.1943 | 0.1335 | 0.1897 | 0.1929 | 0.1280 | 0.1691 | 0.0189 | 0.0078 | 0.0134 | 0.0085 | 0.0189 | 0.0146 |
| Average F1 | 0.354 | 0.227 | 0.343 | 0.358 | 0.207 | 0.355 | 0.015 | 0.018 | 0.013 | 0.013 | 0.015 | 0.016 |
| Best Epoch | 97 | 149 | 149 | 197 | 77 | 151 | 200 | 200 | 197 | 200 | 191 | 200 |

| Backbone | SegFormer-B2 | | | | | | Swin-B | | | | | |
|---|---|---|---|---|---|---|---|---|---|---|---|---|
| Feature Extractor | None | Bayar | DCT | FFT | Sobel | SRM | None | Bayar | DCT | FFT | Sobel | SRM |
| CASIA v1 | 0.6544 | 0.2607 | 0.619 | 0.6187 | 0.2239 | 0.6019 | 0.6831 | 0.4186 | 0.7092 | 0.7147 | 0.3394 | 0.6831 |
| Columbia | 0.8575 | 0.09744 | 0.5571 | 0.588 | 0.1454 | 0.5179 | 0.823 | 0.2159 | 0.6979 | 0.6705 | 0.302 | 0.5143 |
| NIST16 | 0.2767 | 0.1764 | 0.2729 | 0.2487 | 0.2005 | 0.2388 | 0.3133 | 0.1513 | 0.3048 | 0.28 | 0.2255 | 0.2858 |
| Coverage | 0.2711 | 0.107 | 0.1655 | 0.1663 | 0.0821 | 0.2293 | 0.5244 | 0.1493 | 0.324 | 0.2809 | 0.1211 | 0.2416 |
| IMD2020 | 0.2679 | 0.0754 | 0.2042 | 0.1997 | 0.0844 | 0.2248 | 0.3459 | 0.1363 | 0.3223 | 0.2899 | 0.1387 | 0.2494 |
| Average F1 | 0.466 | 0.143 | 0.364 | 0.364 | 0.147 | 0.363 | 0.538 | 0.214 | 0.472 | 0.447 | 0.225 | 0.395 |
| Best Epoch | 151 | 181 | 200 | 169 | 165 | 151 | 77 | 177 | 137 | 157 | 173 | 165 |

| Backbone | ViT-B/16 | | | | | | ViT-B/16-8 | | | | | |
|---|---|---|---|---|---|---|---|---|---|---|---|---|
| Feature Extractor | None | Bayar | DCT | FFT | Sobel | SRM | None | Bayar | DCT | FFT | Sobel | SRM |
| CASIA v1 | 0.5169 | 0.0698 | 0.4001 | 0.4209 | 0.0311 | 0.4633 | 0.4195 | 0.2338 | 0.2989 | 0.2949 | 0.2031 | 0.4308 |
| Columbia | 0.3609 | 0.0854 | 0.2721 | 0.2757 | 0.0524 | 0.3057 | 0.1890 | 0.1441 | 0.1275 | 0.2656 | 0.1747 | 0.2063 |
| NIST16 | 0.2344 | 0.1104 | 0.1944 | 0.2126 | 0.0861 | 0.1943 | 0.2047 | 0.1415 | 0.1560 | 0.1630 | 0.1273 | 0.1932 |
| Coverage | 0.1365 | 0.0463 | 0.1076 | 0.1097 | 0.0256 | 0.1303 | 0.0932 | 0.0585 | 0.0640 | 0.1028 | 0.0614 | 0.0824 |
| IMD2020 | 0.2259 | 0.0284 | 0.1669 | 0.1877 | 0.0144 | 0.1755 | 0.1588 | 0.0869 | 0.1072 | 0.1149 | 0.0711 | 0.1687 |
| Average F1 | 0.295 | 0.068 | 0.228 | 0.241 | 0.042 | 0.254 | 0.213 | 0.133 | 0.151 | 0.188 | 0.128 | 0.216 |
| Best Epoch | 151 | 200 | 191 | 200 | 200 | 184 | 157 | 151 | 200 | 149 | 177 | 165 |

### A.7.2   Modules Detailed Evaluation

As shown in Table 13, except for U-Net, which is completely underfitting, the performance of all backbone models combined with Bayar and Sobel has decreased, indicating that Bayar and Sobel are not suitable for the IMDL tasks. While DCT, FFT, and SRM improve some models, they also reduce performance in others, necessitating an analysis of their combined models' performance.

Regarding backbones, SegFormerB2 and Swin-B are more suitable for manipulation detection tasks than ViT and ResNet. Unet's highest F1 score is close to 200 epochs, indicating that it is seriously underfitting, and the highest indicator is still very poor compared to other models, making it difficult to train and unsuitable for manipulation detection tasks. Comparing the performance of the ViT whole transformer block and the first eight blocks, it can be concluded that the larger ViT model is more suitable.

In terms of model combinations, DCT, FFT, and SRM significantly improve the average F1 score of ResNet151. Compared to ViT's performance without feature extractors, using feature extractors severely slowed down its convergence. However, in the shallow model features fusion technique using feature map concatenation, ViT-B/16-8 converges faster than when modifying the input patch embed layer. Therefore, other feature fusion methods should be considered to determine whether low-level features are effective. When Swin uses feature extractors, its performance on the test set CASIAv1 improves, but performance on other datasets decreases, indicating overfitting. SegFormer is not suitable for all feature extractors.

### A.7.3   Details of the NIST16 Dataset Cleansing Process

Based on the label leakage present in NIST16, we proposed a method to remove similar images from the NIST16 dataset. This ensures that the remaining images do not exhibit overly similar patterns, allowing models to focus on detecting manipulation traces rather than exploiting image patterns to cheat.

We use the SSIM (Structural Similarity) index to determine if two images are similar. NIST16 contains 564 manipulated images, and by calculating the SSIM value for each pair of images, we obtain a 564x564 matrix. Through multiple experiments, we set the threshold for determining whether two images are similar at an SSIM value of 0.9. If the SSIM value exceeds the threshold, we consider

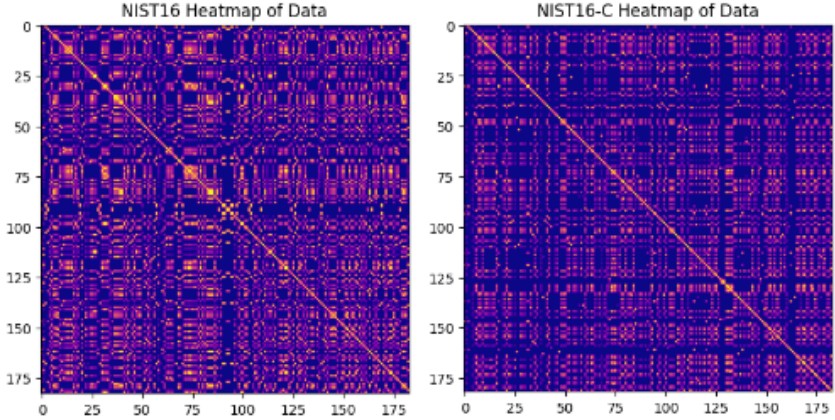

Figure 6: **The heatmaps for NIST16 and NIST16-C.** The brighter the color, the higher the SSIM value at the corresponding position. It can be seen that the heatmap for the NIST16 dataset is noticeably brighter compared to NIST16-C. For NIST16, we randomly selected 183 images to calculate their SSIM values.

Table 14: **Pixel-level performance on the NIST16 dataset.** The pixel-level performance on the NIST16 dataset shows significant differences depending on whether fine-tuning was performed. After fine-tuning, the pixel-level performance shows a substantial increase. For SPAN, the values in parentheses are from SPAN's pre-training."$\times$" means there is no test data.

| Protocol | PSCCNet[25] | Objectformer[40] | SPAN[18] |
|---|---|---|---|
| fine-tuned | 0.742 | 0.826 | 0.582(0.29) |
| Protocol-MVSS | 0.2141 | 0.1732 | $\times$ |
| Protocol-CAT | 0.3689 | 0.2682 | $\times$ |

the two images similar and set the corresponding value in the matrix to 1. Conversely, if the SSIM value is below the threshold, we consider the images dissimilar and set the corresponding value to 0.

We then compute the transitive closure of the resulting matrix and subsequently calculate the connected components of the transitive closure matrix. The images within a single connected component are considered to be from the same source. We performed manual filtering and further divided these source images into multiple groups based on their manipulation methods. Finally, for each group of similar images, we retained only one image that best represents the authentic manipulation.

Through this cleaning process, the NIST16 dataset was reduced to 183 images that do not exhibit label leakage. We named this cleaned dataset NIST16-C. In Figure 6, we compare the heatmaps of NIST16 and NIST16-C. Considering the label leakage issue present in the NIST16 dataset, we propose a method for dividing the NIST16 dataset into training and validation sets. We ensure that similar images are either all in the validation set or all in the training set, thereby increasing the difficulty of training. This eliminates the abnormally high metrics and performance that were previously achieved with random splitting of the training and validation sets on the NIST16 dataset.

### A.7.4 Controversial F1 Metrics

In 2018, Minyoung Huh et al[20]. proposed a permuted metrics approach. They compared the F1 score of the predicted image with the F1 score of the pixel-inverted predicted image, selecting the larger value as the result. The formula is as follows:

$$\text{F1}_{permute}(G, P) = \max\left(\text{F1}(G, P), \text{F1}\left(G, P^C\right)\right) \tag{1}$$

Where $G$ refers to ground truth and $P$ refers to prediction. Such a formula is controversial as it does not accurately reflect the model's detection capabilities. In Figure 4, the model's prediction results are completely incorrect, yet its F1 score is relatively high.

The F1 score of a completely incorrect predicted image should theoretically be 0.00. However, when the white area in the mask exceeds 50%, the F1 score of the inverted image can even reach above 0.66, which is clearly unreasonable. The state-of-the-art models CAT-Net and TruFor are also using this metric. Permuted metrics seem to artificially inflate the scores, suggesting a need for a more reasonable evaluation metric.

In the Python Skearn library, the F1 score function[9] has five options for calculation methods: 'micro', 'macro', 'samples', 'weighted', 'binary'. Except for the 'binary' is the commonly known F1, others are mainly used for **multi-label classification** problems and unsuitable for our tampering detection task with a single binary mask. This means that F1 scores are computed for each class and then averaged. Precisely, when used for binary classification, an additional F1 score is calculated by reversing both the predicted mask and the ground truth, and this score is averaged with the original F1 score. The specific averaging algorithms are different between macro and micro, and the mathematical formulas are as follows:

$$F1 = 2 \times \frac{TP}{2 \times TP + FP + FN}$$

$$F1' = 2 \times \frac{TN}{2 \times TN + FN + FP}$$

$$F1_{macro} = \frac{F1 + F1'}{2} = \frac{TP}{2 \times TP + FP + FN} + \frac{TN}{2 \times TN + FN + FP}$$

$$F1 - F1_{macro} = \frac{TP}{2 \times TP + FP + FN} - \frac{TN}{2 \times TN + FN + FP}$$

$$= \frac{1}{2 + \frac{FP + FN}{TP}} - \frac{1}{2 + \frac{FN + FD}{TN}}$$

$$F1 = 2 \times \frac{TP}{2 \times TP + FP + FN}$$

$$F1_{micro} = \frac{TP + TN}{2 \times (TP + TN) + 2 \times (FP + FN)}$$

$$F1 - F1_{micro} = 2 \times \frac{TP}{2 \times TP + FP + FN} - \frac{TP + TN}{2 \times (TP + TN) + 2 \times (FP + FN)}$$

$$= \frac{(TP - TN) \times (FP + FN)}{(2 \times TP + FP + FN)[2 \times (TP + TN) + 2 \times (FP + FN)]}$$

Since in IMDL the majority of cases typically have $TP << TN$, it follows that $F1 - F1_{macro} < 0$ and $F1 - F1_{micro} < 0$. Consequently, $F1 < F1_{macro}$ and $F1 < F1_{micro}$, leading to an erroneous estimation of model performance.

We have also provided code to verify the specific implementation of macro-F1 and micro-F1 here: https://github.com/scu-zjz/IMDLBenCo/blob/main/tests/test_sklearn_F1s.py

For the example in Figure 4, where the theoretical F1 score is 0.00, the F1 values calculated using the binary, micro, macro, and weighted parameters are 0.00, 0.74, 0.42, and 0.74, respectively. Clearly, except for the binary parameter, all other parameters seem to artificially inflate the F1 score. This is evidently unreasonable and cannot be used to measure the detection accuracy of an image manipulation detection model. However, some studies currently use an unreasonable F1 calculation formula, such as HiFi-Net[14].

### A.7.5   AUC Metrics

**Overoptimistic AUC Value.** AUC is the other widely adopted metric in IMDL. Advanced models with high AUC values may also perform poorly in tamper localization, suggesting that AUC alone does not reliably gauge a model's detection capability. The specific explanation is as follows:

---

[9]https://scikit-learn.org/stable/modules/generated/sklearn.metrics.f1_score.html#sklearn.metrics.f1_score

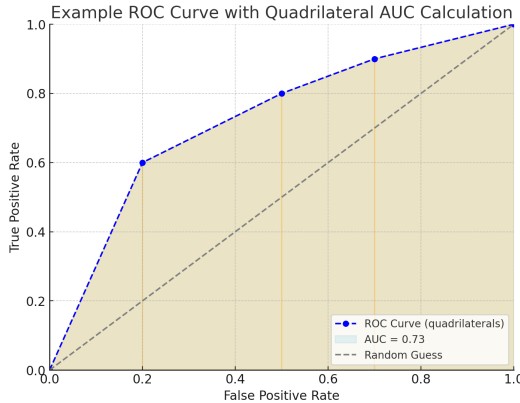

Figure 7: Exapmple of AUC, and how trapezoid constructed when only have three data points.

The Area Under the ROC Curve (AUC) is a metric used to measure the overall performance of a binary classification model across all possible classification thresholds[19]. The ROC curve itself plots the True Positive Rate (TPR) against the False Positive Rate (FPR) at various threshold settings. Visualization is shown in Figure 7.

Because of the threshold values, TPR and FPR are estimated through discrete points in practice. To compute the integral, we leverage the classic trapezoid rule, which involves dividing the area under the ROC curve into as many as possible trapezoids and summing their areas.

Set tuples $(FPR_1, TPR_1), (FPR_2, TPR_2), ..., (FPR_n, TPR_n)$ to represent all the trapezoids

for areas under ROC, and $FPR_i$ is the i-th trapezoids along with a certain threshold value.

Then apply the trapezoid rule:

$$AUC = \int_0^1 f(x)dx \tag{2}$$

$$\approx \sum_{i=1}^{n} \frac{TPR_{i+1} - TPR_i}{2} \times (FPR_{i+1} - FPR_i) \tag{3}$$

$$= \sum_{i=1}^{n} \frac{1}{2}(\frac{TP_{i+1}}{TP_{i+1} + FN_{i+1}}) \times (\frac{FP_{i+1}}{FP_{i+1}TN_{i+1}} - \frac{FP_i}{FP_i + TN_i}) \tag{4}$$

where i-th is the threshold for separating positive and negative samples.

We found that the AUC is calculated by accumulating the results across all thresholds. However, in practice, since the distribution of the dataset is unknown, a threshold of 0.5 is typically chosen for making predictions. This leads to integrating over unnecessary regions when the model's ROC curve deviates significantly from 0.5, resulting in an overestimation. Such a situation is also supported by previous work (Fig.8 in MVSS-Net++[7]).

Further, the manipulated regions in an image are usually very small. Therefore, in manipulated images, the number of manipulated pixels and the number of authentic pixels are extremely unbalanced. Due to such a large skew in class distribution, although AUC performs well on many unbalanced classification problems, it still tends to provide an over-optimistic estimate of IML model performance. Recent studies [4, 48] have also noticed similar issues with finding the optimal threshold, but they did not provide a profound solution. In this paper, we speculate that this disastrous performance is due to overfitting caused by large-scale pre-training, and the AUC metric is not sufficient to address this overfitting problem. Therefore, considering the overly optimistic estimates of the AUC metric in IML, we advocate using only the F1 score to evaluate IML models.

