# OpenReview forum: "IMDL-BenCo: A Comprehensive Benchmark and Codebase for Image Manipulation Detection & Localization"
_NeurIPS.cc/2024/Datasets_and_Benchmarks_Track — NeurIPS 2024 Track Datasets and Benchmarks Spotlight_

### Official Review · Reviewer_21JT · 2024-07-10
**Comments of IMDL-BenCo**

**Rating:** 7
**Confidence:** 5
**Correctness:** I do not see any correctness problems…
**Clarity:** This paper is well-written.

**Review:**

Overall, the contributions of this work are timely and significant for the forensics community, facilitating future advancements in image forgery detection and localization models for researchers. The paper is well-written and well-organized, offering insightful findings and raising important open questions. However, there are some limitations, including the choice of benchmark datasets, evaluation metrics, robustness evaluations, analysis of feature extractors, and the visualization codebase.

**Strengths:**

1. This work presents a standardized process for image manipulation detection and localization. The codebase incorporates four key components: a data loader, a model zoo, a training script, and an evaluator, enhancing the model customization and coding efficiency.

2. This benchmark comprehensively implements eight state-of-the-art models, two training protocols, 15 GPU-accelerated metrics, and three types of robustness evaluations.

3. The paper provides insightful discussions and analyses, offering new insights on IMDL model architecture, dataset characteristics, and evaluation standards.

**Additional Feedback:**

N.A.

**Documentation:**

No dacumentation issue found.

**Ethics:**

I do not find any obvious and serious ethic problems in this paper.

**Limitations:**

1. Benchmark Datasets: The presented benchmark, while comprehensive, includes only four datasets for testing: Coverage, Columbia, NIST16, and CASIAv1. To better demonstrate the cross-dataset generalization capability of existing detectors, more testing datasets should be included in both pixel-level and image-level evaluations.

2. Evaluation Metrics: This work mainly reports the F1-score for pixel-level and image-level forgery detection. The authors point out that the F1-score does not fully represent the model’s forgery detection and localization capability. Including metrics such as IoU and AUC in the experimental results would provide a more comprehensive evaluation of detector performance. IoU can naturally avoid the issue presented in Fig. 4, and AUC, being a threshold-free metric, can objectively reflect model performance. The combination of F1, IoU, and AUC would better represent an image forgery detection model’s capabilities.

3. Robustness Evaluations: While this work conducts robustness evaluations on three perturbation types: JPEG compression, Gaussian blur, and Gaussian noise, more perturbations, such as contrast and illumination variations, are expected to comprehensively evaluate the robustness of existing models in real-world applications.

4. Analysis of Feature Extractors: The benchmark found that Bayar and Sobel feature extractors are not suitable for IMDL, which is an interesting finding. More analysis and discussion are expected to better interpret this observation.

5. Visualization Codebase: Since image forgery localization maps are important in the IMDL task, it would be beneficial to incorporate standardized image forgery visualization modules into the codebase.

6. Equation (2) in Appendix B.5.5: Equation (2) appears incorrect. AUC calculates the area under the ROC curve; however, the equation suggests that AUC is proportional to the product of TPR and (1-FPR), which is unreasonable.

**Opportunities For Improvement:**

See limitations.

**Relation To Prior Work:**

Yes, the author have shown a clear comparison to compare with previous works.

**Summary And Contributions:**

This work introduces a standardized and user-friendly benchmark for image manipulation detection and localization. The developed codebase includes four key components: eight state-of-the-art IMDL models, two sets of standard training and evaluation protocols, 15 GPU-accelerated evaluation metrics, and three types of robustness evaluations.

---

> ### Author Response · Authors · 2024-08-13
> **We need more time on experiments.**
>
> Thanks for your insightful comments. We are completing extra experiments on evaluating all models with AUC and IoU metrics, and will respond to your comments once our experiments are done.

---

> > ### Comment · Reviewer_21JT · 2024-08-30
> >
> > Thanks for the authors detailed responses and the additional experiments. I have promoted the rating from 6 to 7. I suggest the authors just use 'the area under ROC curve' to define AUC, instead of using equations.

---

> > > ### Author Response · Authors · 2024-08-30
> > >
> > > Thank you for accurately identifying the errors in our paper—this is the most valuable feedback we could receive. We will be cautious with the term 'Area under the ROC curve' instead of equations for greater precision. We also sincerely appreciate your positive feedback on our rebuttal.

---

> ### Author Rebuttal · Authors · 2024-08-23
>
> Thank you for your valuable and comprehensive feedback and your thoughtful comments on our manuscript. We appreciate the time and effort you have invested in reviewing our work. Here is our  response to address your concerns.
>
> ****
>
> **Q1: More Benchmark Datasets:**
> - A1:
>   - Currently, there are relatively few effective datasets in the field of tampering detection. In our papar, we have utilized CASIAv2, Coverage, Columbia, NIST16 and IMD2020 dataset for experiments, which are all high-quality, well known benchmark datasets. Current mainstream state-of-the-art work also largely focuses on these datasets.
>   - Specifically, the Defacto dataset was not included in this study due to its poor performance in Protocol-MVSS, with an F1 score of less than 0.15. This score is approximately equivalent to predicting everything as white, rendering it meaningless as a reference.
>   - There are also other datasets that lack publicly available access or whose quality has not been accepted by mainstream work, so they are not included for now. We will continue to acquire and update these datasets in the future.
>
> **Q2: More Evaluation Metrics:**
> - A2:
>   - Thanks for pointing out the needs of including more metrics. We believe that IoU is indeed a good metric, even though it is not widely used in the IML field at present, it can effectively reflect performance. Additionally, AUC is also valuable; however, many studies have mixed the use of fixed and optimal AUC, leading to unfair comparisons. **The results are shown below and will be included in the Appendix.** If better metrics become available in the future, we will also add them into codebase.
> - Protocol-MVSS-AUC:
>   - | Method       |  COVERAGE     | Columbia | NIST16 | CASIAv1 | IMD2020 | Average |
> |--------------|---------------|----------|--------|---------|---------|---------|
> | MVSS-Net     | 0.720         | 0.732    | 0.737  | 0.861   | 0.755   | 0.761   |
> | CAT-Net      | 0.759         | 0.800    | 0.787  | 0.910   | 0.775   | 0.806   |
> | Objectformer | 0.739         | 0.528    | 0.722  | 0.876   | 0.680   | 0.709   |
> | PSCC-Net     | 0.697         | 0.814    | 0.725  | 0.833   | 0.778   | 0.769   |
> | Trufor       | 0.912         | 0.928    | 0.820  | 0.946   | 0.840   | 0.889   |
> | IML-ViT      | 0.871         | 0.898    | 0.789  | 0.940   | 0.814   | 0.862   |
> - Protocol-MVSS-IOU
>   - | Method       |  COVERAGE     | Columbia | NIST16 | CASIAv1 | IMD2020 | Average  |
> |--------------|---------------|----------|--------|---------|---------|----------|
> | MVSS-Net     | 0.192         | 0.290    | 0.176  | 0.440   | 0.201   | 0.260    |
> | CAT-Net      | 0.204         | 0.469    | 0.196  | 0.509   | 0.198   | 0.315    |
> | Objectformer | 0.181         | 0.224    | 0.111  | 0.320   | 0.106   | 0.189    |
> | PSCC-Net     | 0.146         | 0.477    | 0.150  | 0.310   | 0.157   | 0.248    |
> | Trufor       | 0.352         | 0.833    | 0.271  | 0.666   | 0.267   | 0.478    |
> | IML-ViT      | 0.372         | 0.685    | 0.250  | 0.648   | 0.264   | 0.444    |
> - Protocol-CAT-AUC:
>   - | Method       |  COVERAGE     | Columbia | NIST16 | CASIAv1 | IMD2020 | Average  |
> |--------------|---------------|----------|--------|---------|---------|----------|
> | MVSS-Net     | 0.870         | 0.933    | 0.790  | 0.912   | -       | 0.876    |
> | CAT-Net      | 0.917         | 0.946    | 0.822  | 0.980   | -       | 0.916    |
> | Objectformer | 0.747         | 0.858    | 0.775  | 0.884   | -       | 0.816    |
> | PSCC-Net     | 0.884         | 0.946    | 0.828  | 0.919   | -       | 0.894    |
> | Trufor       | 0.942         | 0.899    | 0.878  | 0.974   | -       | 0.924    |
> | IML-ViT      | 0.942         | 0.955    | 0.893  | 0.976   | -       | 0.942    |
> - Protocol-CAT-IOU:
>   - | Method       |  COVERAGE     | Columbia | NIST16 | CASIAv1 | IMD2020 | Average  |
> |--------------|---------------|----------|--------|---------|---------|----------|
> | MVSS-Net     | 0.389         | 0.672    | 0.259  | 0.491   | -       | 0.453    |
> | CAT-Net      | 0.387         | 0.895    | 0.212  | 0.748   | -       | 0.561    |
> | Objectformer | 0.188         | 0.670    | 0.203  | 0.320   | -       | 0.345    |
> | PSCC-Net     | 0.301         | 0.814    | 0.297  | 0.500   | -       | 0.478    |
> | Trufor       | 0.415         | 0.859    | 0.296  | 0.775   | -       | 0.586    |
> | IML-ViT      | 0.590         | 0.933    | 0.440  | 0.749   | -       | 0.678    |
>
> **Q3: More Robustness Evaluations**
> - In the real world, tampering detection tasks indeed face many types of perturbations. However, considering experimental costs and other factors, current mainstream IML models typically only address these three types of perturbations, which is a widely accepted consensus. Therefore, to stand in line with the original works, we can only report results based on this standard in the paper. However, in fact, the implementation of IMDL-BenCo's code can further accommodate almost any perturbation type available in the Albumentations library.
> - We will gradually add the officially provided perturbations to benco as well.

---

> > ### Author Rebuttal · Authors · 2024-08-23
> >
> > **Q4: Analysis of Feature Extractors:**
> > - A4:
> >   - We tend to answer this question 3-fold:
> >   - First, we believe that an essential factor is that these two feature extractors are not as universally applicable as initially thought. They may only be capable of extracting features from specific patterns, and they may also exhibit poor resistance to resizing.
> >   - Second, this paper only explores shallow fusion methods, specifically by concatenating the feature extractor with the original RGB image at the first layer. This approach is more suited for CNNs, as it enables gradual fusion. However, for transformers, it creates a bottleneck by placing almost all fusion tasks on the embedding layer.
> >   - While this does not imply that deeper fusion methods will necessarily perform better, practice is the ultimate test of truth. Therefore, further experimentation is required. However, due to space constraints, this paper cannot fully investigate this issue, such as testing on various resolutions or exploring deeper feature fusion methods. Nevertheless, the IMDL-Benco framework provides an efficient way to conduct relevant ablation experiments and analyses for future studies.
> >
> > **Q5: Visualization Codebase**
> > - A5:
> >   - Thanks for point out this requirements. As shown in line 128, the IMDL-Benco has provided the interface for visualizing intermediate features. Specifically, you can check our implementation in TruFor as an example: https://github.com/scu-zjz/IMDLBenCo/blob/abd811a4408ca2c2bcec73004012e82c23dc2f37/IMDLBenCo/model_zoo/trufor/trufor.py#L100. You can pass all kinds of tensors into the `visual_image` dictionary, then the IMDL-BenCo will automaticly visualize this feature in logging with Tensorboard.
> >   - Thus, the feature you requested is actually already implemented in the  IMDL-BenCo. If you have further suggestions on the interface, please feel free to discuss them with comments.

---

> > ### Author Rebuttal · Authors · 2024-08-24
> >
> > **Q6: Equation (2) in Appendix B.5.5: Equation (2) appears incorrect.**
> > - A6:
> >   - We examined this equation, and we **do miss a sum notation in it**. Thanks very much for your careful review. The correct derivation process should be:
> > $$
> > AUC=\int_{0}^{1}f(x)\ dx \tag{1}
> > $$
> > where $f(x)$ is the ROC curve
> >
> > Also, the ROC curve is built upon TPR (True Positive Rate) & FPR (False Positive Rate), where TPR is a function of FPR. And with different thresholds, we have various TPR & FPR ranging from [0,1].
> >
> > Because of the threshold values, in practice, TPR and FPR are estimated through discrete points. To compute the integral, we leverage the classic trapezoid rule, which involves dividing the area under the ROC curve into as many as possible trapezoids and summing their areas.
> >
> > Set tuples $(FPR_1, TPR_1), (FPR_2, TPR_2), ... , (FPR_n, TPR_n)$ to represent all the trapezoids
> >
> > for areas under ROC, and FPR_i is the i-th trapezoids along with a certain threshold value.
> >
> > Then apply the trapezoid rule:
> > $$
> > \begin{align}
> > AUC&=\int_0^1f(x)dx \\\\
> > &\approx \sum_{i=1}^{n}\frac{TPR_{i+1} - TPR_i}{2} \times (FPR_{i+1} - FPR_i) \\\\
> > &= \sum_{i=1}^{n}\frac{1}{2} (\frac{TP_{i+1}}{TP_{i+1}+FN_{i+1}}) \times (\frac{FP_{i+1}}{FP_{i+1}TN_{i+1}}- \frac{FP_i}{FP_i + TN_i}) \tag{2}
> > \end{align}
> > $$
> > where i-th is the threshold for separating positive and negative samples.
> >
> > **First, Equation 2 correctly reflects the formula we mistakenly wrote in the Appendix, which we will correct in the final version.**
> >
> > Second, we found that the AUC is calculated by accumulating the results across all thresholds. However, in practice, since the distribution of the dataset is unknown, a threshold of 0.5 is typically chosen for making predictions. This leads to integrating over unnecessary regions when the model's ROC curve deviates significantly from 0.5, resulting in an overestimation. Such a situation is also supported by previous work (Fig.8 in MVSS-Net++[A]).
> >
> > We will correct this equation and add the derivation process to our manuscript. And sincerely thanks again for your careful reviewing& valuable suggestions.
> >
> > ## Reference
> > Dong, C., Chen, X., Hu, R., Cao, J., & Li, X. (2022). Mvss-net: Multi-view multi-scale supervised networks for image manipulation detection. IEEE Transactions on Pattern Analysis and Machine Intelligence, 45(3), 3539-3553.

---

> ### Author Response · Authors · 2024-08-28
>
> Dear Reviewer 21JT,
>
> Thank you for your thorough evaluation of our work. We are committed to incorporating your feedback comprehensively into our revised paper to improve both its content and overall quality.
>
> As the discussion period draws to a close, we hope our response has sufficiently addressed your concerns. If there are any additional issues or points that require further clarification, we are more than willing to address them promptly.
>
> Best regards, The Authors

---

### Official Review · Reviewer_iyN9 · 2024-07-24

**Rating:** 6
**Confidence:** 2
**Correctness:** yes
**Clarity:** yes

**Review:**

Strengths:
1. Comprehensive Benchmark: The introduction of a unified benchmark and codebase for IMDL addresses a critical need in the field. This will help standardize evaluations and facilitate fair comparisons among different IMDL models, even though this area is very niche.
2. Modular Codebase: The modularity allows researchers to customize and extend components easily.
3. Open Source Availability: Making the code available on GitHub promotes transparency and encourages further research and development in the IMDL field.

Weaknesses:
1. Implementation Details: While the paper provides a general overview of the codebase and its components, more detailed implementation specifics, such as code snippets or configuration examples, would be beneficial for reproducibility.
2. Correctness of Results: With eight authors contributing equally and each responsible for one core method, there is a concern about the consistency and fairness in the implementation and evaluation of each part. This necessitates further verification from more researchers.

Suggestions:
1. Detailed Implementation Examples: Include more detailed examples in the appendix or supplementary materials to aid researchers in replicating and extending the work.
2. Discussion on Performance Variations: Provide a more in-depth discussion on the performance variations observed in the reproduced models, including potential causes and mitigation strategies.

**Strengths:**

1. Comprehensive Benchmark: The introduction of a unified benchmark and codebase for IMDL addresses a critical need in the field. This will help standardize evaluations and facilitate fair comparisons among different IMDL models, even though this area is very niche.
2. Modular Codebase: The modularity allows researchers to customize and extend components easily.
3. Open Source Availability: Making the code available on GitHub promotes transparency and encourages further research and development in the IMDL field.

**Additional Feedback:**

no

**Documentation:**

yes

**Opportunities For Improvement:**

Weaknesses:
1. Implementation Details: While the paper provides a general overview of the codebase and its components, more detailed implementation specifics, such as code snippets or configuration examples, would be beneficial for reproducibility.
2. Correctness of Results: With eight authors contributing equally and each responsible for one core method, there is a concern about the consistency and fairness in the implementation and evaluation of each part. This necessitates further verification from more researchers.

Suggestions:
1. Detailed Implementation Examples: Include more detailed examples in the appendix or supplementary materials to aid researchers in replicating and extending the work.
2. Discussion on Performance Variations: Provide a more in-depth discussion on the performance variations observed in the reproduced models, including potential causes and mitigation strategies.

**Relation To Prior Work:**

yes

**Summary And Contributions:**

The paper presents IMDL-BenCo, a comprehensive benchmark and modular codebase for Image Manipulation Detection & Localization (IMDL). IMDL-BenCo aims to provide a unified framework that includes standardized components, training protocols, and evaluation metrics, facilitating rigorous experimentation and fair comparisons among IMDL models. The codebase includes implementations of eight IMDL models.

---

> ### Author Rebuttal · Authors · 2024-08-12
>
> Thank you for your valuable feedback and thoughtful comments on our manuscript. We appreciate the time and effort you have invested in reviewing our work. Here is our response to address your concerns.
>
> ****
>
> **Q1:Implementation Details: While the paper provides a general overview of the codebase and its components, more detailed implementation specifics, such as code snippets or configuration examples, would be beneficial for reproducibility.**
> - A1:
>   - Thank you for your valuable suggestions. But as a code framework that will be maintained long-term, we can not guarantee that all details will remain "fixed." Therefore, we believe that including specific code snippets in the paper may not be appropriate, as they are likely to become outdated. To facilitate understanding and usage for researchers and developers, we will present this information through documentation (https://scu-zjz.github.io/IMDLBenCo-doc/) and comments in the code. We will focus on advancing the documentation after the Rebuttal phase is completed.
>
> **Q2:Correctness of Results: With eight authors contributing equally and each responsible for one core method, there is a concern about the consistency and fairness in the implementation and evaluation of each part. This necessitates further verification from more researchers.**
> - A1:
>   - First, I need kindly point out that our `author contributins` is located at the Appendix A (Line 447), with only five authors are responsible for reproducing core methods, not eight people. The first author, Xiaochen Ma, is responsible for overseeing all implementations and hyperparameter tuning to ensure that the parameter settings are as faithful as possible to the original paper, with no errors in implementation details. Additionally, all models are trained multiple times to explore optimal settings and details (since our GPU may differ from the original paper), and reliable logs and checkpoints are maintained for check. Thus, we have the confidence that we are doing our best to maintain the consistency across all models in the `model_zoo`. We also welcome industry and researchers' scrutiny of all implementation details and are open to supervision and criticism. Furthermore, we will release the best-performing checkpoints on GitHub to facilitate direct testing of each model's performance.
>   - While we are indeed willing to conduct a detailed analysis of the performance fluctuations of each model, it is difficult to draw conclusions without experiments. This requires a thorough comparative experimental design and training analysis for each model. Given the constraints on the paper's length, we prefer to include this analysis in the subsequent documentation (https://scu-zjz.github.io/IMDLBenCo-doc/) for a more comprehensive analysis.

---

> ### Author Rebuttal · Authors · 2024-08-20
>
> # 2nd response for Reviewer iyN9
>
> Dear Reviewer iyN9
> Thank you for your thoughtful and constructive feedback on our manuscript. As the authors of a codebase and a benchmark for Image Manipulation Detection & Localization (IMDL), we fully understand that the readability and correctness of our work are critical in deciding the usability and the overall application of our research. Therefore, we greatly value your suggestions and are further committed to devising a detailed schedule below to improve the readability and correctness of our codebase.
>
> ## Schedule for codebase improvement:
>
> ### **For Detailed Implementation Examples**
>
> We plan to include detailed implementation examples in the project documents and user guidance to aid researchers in replicating and extending the work. These examples will provide step-by-step guidance on how to set up and run experiments using our codebase. This will be implemented through:
> 1. Documentation Expansion:
>   - Enhance the documentation by adding detailed implementation examples for each model included in the codebase. These examples will cover setup instructions, configuration parameters, and code snippets that illustrate how to use specific features or functionalities within the codebase.
>   - Create a dedicated section in the documentation for each model (extra comments), detailing the specific configurations used in the experiments reported in the paper. This can include hyperparameters, pre-processing steps, and any other relevant settings.
> 2. Supplementary Materials:
>   - Include additional supplementary materials such as Jupyter notebooks or scripts that replicate key results presented in the paper. These materials will be self-contained and easy to execute, allowing users to quickly verify the reported findings.
>   - Publish these supplementary materials alongside the main repository or in a separate repository linked from the main project page. Ensure that they are well-organized and clearly labeled with instructions for use.
> 3. Code Snippets:
> - Integrate code snippets into the documentation that demonstrate how to use various components of the codebase. These snippets will highlight important aspects of the code, making it easier for users to understand the underlying logic and functionality.
> 4. Version Control:
> - Use version control systems effectively to track changes in the documentation and supplementary materials. This helps maintain a history of updates and ensures that all versions are accessible for reproducibility purposes.
>
> ### **For Discussion on Performance Variations**
> - Regarding the performance variations observed in the reproduced models, we agree that a more in-depth discussion would be beneficial. While we are indeed willing to conduct a detailed analysis of the performance fluctuations of each model, it is difficult to draw conclusions without experiments. This requires a thorough comparative experimental design and training analysis for each model. Given the constraints on the paper's length, we prefer to include this analysis in the subsequent documentation (https://scu-zjz.github.io/IMDLBenCo-doc/) for a more comprehensive analysis. We will provide insights into potential causes of performance variations and suggest mitigation strategies to help future researchers understand and optimize the models more effectively. This will be implemented through:
>
> 1. Documentation of Variability Factors:
>   - Document the potential factors that contribute to performance variability in the models. This documentation should include discussions on data quality, pre-processing techniques, feature extraction methods, and model architecture choices.
> Provide guidelines on how to mitigate the impact of these factors. For example, suggest best practices for data augmentation, recommend specific pre-processing steps, or outline how to tune hyperparameters effectively.
> 2. Establish Standard and ready-to-use Checkpoint
>   - We will release the best-performing checkpoints for each model on GitHub. This will allow other researchers to directly test the performance of our models without having to retrain them from scratch. We also would like to share the training scripts and logs that were used to generate the best-performing checkpoints. This will enable others to replicate the training process if needed.
> 3. Community Engagement:
>    - Encourage community engagement by inviting contributions from external researchers who can validate the results independently. Consider setting up a dedicated issue tracker or forum where users can report discrepancies or suggest improvements.
> Regularly update the documentation and supplementary materials based on feedback received from the community. This ongoing dialogue will help improve the quality and reliability of the benchmark and codebase over time.
>
> By following this schedule and implementing these methods, the authors are confident to enhance the readability and correctness of this benchmark and codebase, thereby addressing your suggestions and improving the overall quality of the project.

---

> ### Author Response · Authors · 2024-08-24
>
> Dear Reviewer iyN9,
>
> Thank you for your thorough evaluation of our work. We are committed to incorporating your feedback comprehensively into our revised paper to improve both its content and overall quality.
>
> As the discussion period draws to a close, we hope our response has sufficiently addressed your concerns. If there are any additional issues or points that require further clarification, we are more than willing to address them promptly.
>
> Best regards,
> The Authors

---

### Official Review · Reviewer_Kx1E · 2024-07-30
**The authors solve some important issues on IMDL field**

**Rating:** 8
**Confidence:** 5

**Review:**

The scarcity of open-sourced baseline models and inconsistent training and evaluation protocols make conducting rigorous experiments and faithful comparisons among IMDL models challenging. For the above issues, the authors present a complete set of evaluation tools. The codebase contains multiple detection algorithms, and the author conducts experimental analysis from various aspects.

**Strengths:**

At present, there is indeed a problem of non-uniform method evaluation in the IMDL field. Specifically, different works may use different training and evaluation strategies. Therefore, the work proposed by the authors, including: the reproduction of undisclosed algorithms, the unification of training scripts and test scripts, etc., is of great significance and can effectively promote the development of the community.

**Additional Feedback:**

Some of the illustrations in the manuscript are rather vague, such as Figure 3, making it difficult to see the details clearly. The author should address this issue.

**Clarity:**

Yes, I think it’s very well written. The language is precise, and the author has done a great job of staying focused on the main topic.

**Correctness:**

Yes. The benchmark is well designed. The evaluation criteria are clear, and the experiments have been conducted in a way that allows for fair and meaningful comparisons.

**Documentation:**

Yes, the author has open-sourced the relevant code and corresponding documentation.

**Ethics:**

No.

**Limitations:**

No, the author should further discuss the limitations of the proposed work for subsequent improvement by other researchers.

**Opportunities For Improvement:**

After looking at the author's open source code, I think the amount of work involved in this endeavor is significant. It might be easier to understand if the author could add more comments to the code.

**Relation To Prior Work:**

Yes.

**Summary And Contributions:**

Currently, Image Manipulation Detection \& Localization (IMDL) field lacks a benchmark to evaluate the performance of detection models. Therefore, the authors propose the first comprehensive IMDL benchmark and modular codebase, termed as IMDL-BenCo.
The proposed IMDL-BenCo include following contributions:
i) decomposes the IMDL framework into standardized, reusable components and revises the model construction pipeline,
improving coding efficiency and customization flexibility;
ii) fully implements or incorporates training code for state-of-the-art models to establish a comprehensive IMDL benchmark;
iii) conducts deep analysis based on the established benchmark and codebase, offering new insights into IMDL model architecture, dataset characteristics, and evaluation standards. Specifically, IMDL-BenCo includes common processing algorithms, 8 state-of-the-art IMDL models (1 of which are reproduced from scratch), 2 sets of standard training and evaluation protocols, 15 GPU-accelerated evaluation metrics, and 3 kinds of robustness evaluation.

---

> ### Author Rebuttal · Authors · 2024-08-11
>
> Thank you for your valuable feedback and thoughtful comments on our manuscript. We appreciate the time and effort you have invested in reviewing our work. Here is our response to further improve our work.
>
> ------
>
> **Q1: It might be easier to understand if the author could add more comments to the code.**
> - A1:
>   - We will focus on improving the documentation after the Rebuttal period. To convince you of further development, we will also insert more comments. Mainly, it helps developers gain a quick understanding of the detailed designing paradigm. We will start this process after the rebuttal period.
>   - We recommend using IMDL-BenCo through a Command Line Interface (CLI), which automatically generates scripts such as train.py for subsequent use and modification. We will focus on improving the comments related to the train and test scripts. For other components, we will enhance understanding through documentation(https://scu-zjz.github.io/IMDLBenCo-doc/).
>
> **Q2: the author should further discuss the limitations of the proposed work for subsequent improvement by other researchers.**
> - A2:
>   - When focusing on the previous methods like DOA-GAN with a GAN training paradigm, the BenCo framework can face a great challenge of integrating the model into the default model_zoo. However, with the support of the CLI paradigm, you can generate the `train.py` and then modify it as you want, including revising the training paradigm. We believe that only re-use modules like augmentation, evaluators and visualizing tools can still greatly help in developing research code.
>
> **Q3: Some of the illustrations in the manuscript are rather vague, such as Figure 3.**
> - A3:
>   - Sorry for this matter; we will improve the DPI of such a Figure to maintain a better reading experience.
>
> If you have further issues, we are glad to discuss them with you sincerely.

---

> ### Author Response · Authors · 2024-08-24
>
> Dear Reviewer Kx1E,
>
> Thank you for your thorough evaluation of our work. We are committed to incorporating your feedback comprehensively into our revised paper to improve both its content and overall quality.
>
> As the discussion period draws to a close, we hope our response has sufficiently addressed your concerns. If there are any additional issues or points that require further clarification, we are more than willing to address them promptly.
>
> Best regards,
> The Authors

---

> ### Comment · Reviewer_Kx1E · 2024-08-29
>
> Thank the authors for their answers to my concerns. I maintain my original score.

---

> > ### Author Response · Authors · 2024-08-29
> >
> > Thank you for your constructive feedback. We will focus on improving the documentation and comments to enhance the overall usability of the framework, making it easier to get started.

---

### Official Review · Reviewer_oatD · 2024-08-02
**Comments on "IMDL-BenCo: A Comprehensive Benchmark and Codebase for Image Manipulation Detection & Localization"**

**Rating:** 9
**Confidence:** 4
**Correctness:** Yes
**Clarity:** Yes

**Review:**

Please see Strengths and Limitations.

**Strengths:**

1. The definition, challenges, and current issues in IMDL are well explained. The high quality writing makes me believe this is a valuable work.

2. According to my reproduce experience in IMDL, I fully agree with the opinion proposed by the authors. It is necessary to provide an open-source codebase in IMDL to ensure a fair comparison with previous work.

3. The analysis in Sec. 5 makes sense to me, and some findings align with my personal experiments.

**Additional Feedback:**

N/A

**Documentation:**

N/A

**Limitations:**

1. From lines 77-90, the inconsistent training issue is pointed out and explained, but the definition of "fairness" is still very hard. We know that different methods needs different training strategy， e.g., learning rate and augmentation. And these method maybe feasible only on some speical pre-trained parameters and basic backbones. Forcing different methods to adopt a specified unified framework may seem fair but is actually unfair. Which parts need to be unified and which do not should be analyzed in depth.

2. Though both CNN and ViT-based backbones are evaluated in this paper, the results show that ViTs achieve much better performance than CNNs. However, the adopted CNNs, such as ResNet and U-Net, are outdated, while the adopted ViTs are comparatively newer. I think some advanced CNNs need to be examined.

3. Low-level feature extractors seem to be not feasible in IMDL, which is contradictory to previous work. Though it is an interesting finding, I think more analysis rather than just results is needed to support it.

**Opportunities For Improvement:**

See the limitations

**Relation To Prior Work:**

Yes

**Summary And Contributions:**

This paper first points out the problem of unfair comparison in previous papers,  including inconsistent training and evaluation protocols, then propose a new evaluation benchmark in an unified framework. The authors also claim that the open accessed codebase is one of the main contribution in this paper. Overall, the motivation, contribution, and deep analysis on previous methods are clear in this paper. I believe this work will contribute to the following works and tend to give a positive socre.

---

> ### Author Response · Authors · 2024-08-13
> **We need some time for experiments.**
>
> Thanks for your insightful comments. We are completing extra experiments on ConvNeXt, a modern CNN architecture, and will respond to your comments once our experiments are done.

---

> ### Author Rebuttal · Authors · 2024-08-23
>
> Thank you for your recognition based on your rich experience and your thoughtful comments on our manuscript. We appreciate the time and effort you have invested in reviewing our work. Here is our  response to address your concerns.
>
> *****
>
> **Q1:  Which parts need to be unified and which do not should be analyzed in depth?**
> - A1:
>   - Thanks for this great question. Honestly, we do consider this issue while designing our code and framework, but in an implicit manner. Showing this design logic in an explicit way is truly helpful. In the following, we explicitly showcase the reasons and parts that need to be unified and should not be analyzed in depth. The principle we base our division on is "model-dependent or model-independent."
>   - **Parts Need to be Unified:**
>     - Training/Test Dataset
>     - Augmentation
>     - Evaluation metrics
>
>     - **Reason**: In any case, these three aspects are model-independent, which must remain unchanged; otherwise, it would directly affect the fairness and impartiality of the experiments.
>    - On the other hand, all model-dependent factors should allow for full customization, as different structures may require different optimal settings. Specifically:
>    - **No need to be Unified:**
>       - All hyper-parameters (Like Epoch, learning rate, momentum of optimizer)
>         - **Reason**: Different model structures (such as CNN or Transformer), loss design (classification or regression), loss magnitude, required epochs for convergence, and downward trends all vary significantly. Consequently, each model has its own set of optimal parameters. In IMDL-BenCo, we adhere as closely as possible to the hyperparameter settings reported in each respective SoTA paper while also retraining and adjusting parameters multiple times when necessary (due to differences in our GPU resources compared to the original papers). Therefore, the hyperparameters for different reproduced models in this paper are quite different, aiming to maximize the performance of each model.  In short, hyperparameters are highly model-customized. They cannot be unified and shall not be unified such that optimal performances on individual models can be ensured to establish fair evaluations.
>       - Resolution of input images.
>         - **Reason1**: Different models are designed with varying default resolutions, balancing reasonable complexity (especially in terms of FLOPs). In most cases, the complexity remains within the same order of magnitude, allowing training to be completed in an acceptable timeframe on several current high-performance GPUs. Abruptly changing the resolution can result in extremely high or low complexity, leading to unfair comparisons between models.
>         - **Reason2**: Many Transformer-based models can only accept inputs of fixed size and cannot freely adjust input resolution without modifying the backbone. To maintain original performance, it is difficult to standardize the resolution.
>       - Model Structure.
>         - **Reason**: This is the basis for comparing the performance of different models.
>   - In fact, our implementation of these aspects is evident in IMDL-Benco’s model design.
>     - To ensure uniformity, we specifically implemented a unified dataset protocol and dataloader to guarantee uniformity across datasets. Additionally, we adopted a consistent augmentation strategy and developed a standardized evaluator, highlighting issues with metrics such as permute-F1 and macro-F1.
>     - For aspects where non-uniformity is intended, we employed different hyperparameters in specific shell scripts (see [GitHub IMDL-BenCo model_zoo](https://github.com/scu-zjz/IMDLBenCo/tree/main/IMDLBenCo/statics/model_zoo/runs)). In Tables 3 and 4, the default resolution for each model was used, and for each model, we implemented the Benco paradigm in the code.
>
> **Q2: I think some advanced CNNs need to be examined.**
> - A2:
>   - Thank you for the reminder. So far, we have primarily tested CNN models that are widely used as backbones in IML tasks. Indeed, the CNN models currently used in the field are somewhat outdated, and there is a need to test more advanced CNNs. Therefore, we included ConvNeXt, a state-of-the-art model, in the tests shown in Table 10. The results are as follows:
>   - |                | None    | Bayar    | DCT      | FFT    | Sobel   | SRM      |
> | -------------- | ------- | -------- | -------- | ------ | ------- | -------- |
> | **CASIA v1**   | 0.7374  | 0.4079   | 0.6626   | 0.6486 | 0.2849  | 0.6564   |
> | **Columbia**   | 0.763   | 0.236    | 0.4782   | 0.5153 | 0.3286  | 0.3722   |
> | **NIST16**     | 0.3388  | 0.1639   | 0.2909   | 0.2896 | 0.2064  | 0.2796   |
> | **Coverage**   | 0.5506  | 0.2183   | 0.3196   | 0.3313 | 0.1407  | 0.1925   |
> | **Best_epoch** | 109     | 161      | 127      | 93     | 93      | 121      |
> | **Average F1** | 0.59745 | 0.256525 | 0.437825 | 0.4462 | 0.24015 | 0.375175 |
>
>   - Thus, we can conclude that the current SoTA IMDL models are relatively weaker compared to the latest SoTA general vision backbones. By leveraging these new vision backbones, whether CNNs or Transformers, it is possible to design stronger and more effective models. In the future, we will also introduce more powerful segmentation-related backbones into Benco.

---

> > ### Author Rebuttal · Authors · 2024-08-23
> >
> > **Q3: Need more analysis on why Low-level feature extractors seem to be not feasible in IMDL.**
> > - A3:
> >   - We have conducted some preliminary analyses; however, as this paper primarily focuses on addressing issues related to the codebase and benchmark, space is limited, so we cannot include all the analysis processes. However, we can address the questions from the following perspectives:
> >     - The performance of the feature extractor is constrained due to the incompatibility between shallow fusion and transformers. This paper only tests **shallow fusion** methods, specifically **concatenating the feature extractor with the original RGB image at the first layer**. This method is relatively more suitable for CNNs, as it allows for gradual fusion, whereas for transformers, it becomes a bottleneck because it imposes almost all the fusion tasks on the embedding layer.
> >     - Of course, this can not be concluded with `Deeper fusion performance better`. The practice is the sole criterion for testing the truth. Thus, further experiments on it are needed. However, Due to limitations in length, this paper cannot fully explore this issue. (Like testing on various resolutions, explore deeper feature fusion methods.) Anyway, using the IMDL-Benco framework can efficiently conduct corresponding ablation experiments and related analyses for future study.
> >     - To some extent, this experiments indicates that some feature extractors may not indeed adapt well to IMDL task. Should raise some attention to the community.
> >   - The issue with the extractor is indeed worth discussing, and we will include this in the manuscript and explore it further in subsequent iterations of Benco.

---

> > > ### Comment · Reviewer_oatD · 2024-08-30
> > >
> > > The authors have addressed my concerns point by point. I also reviewed the comments from the other reviewers and found that all of them appreciate this work. Throughout the rebuttal process, I have observed the authors' efforts to address the reviewers' questions thoroughly, and I believe this paper deserves to be published in NIPS. I hope the authors will continue to update this project on GitHub. Finally, I would like to raise my score to 9, reflecting my strong support for this work.

---

> > > > ### Author Response · Authors · 2024-08-30
> > > >
> > > > Thank you for your invaluable time and positive feedback. We will make every effort to maintain this project on GitHub and contribute to the research community. Your encouragement fuels our passion and commitment.

---

> ### Author Response · Authors · 2024-08-28
>
> Comment:
> Dear Reviewer oatD,
>
> Thank you for your thorough evaluation of our work. We are committed to incorporating your feedback comprehensively into our revised paper to improve both its content and overall quality.
>
> As the discussion period draws to a close, we hope our response has sufficiently addressed your concerns. If there are any additional issues or points that require further clarification, we are more than willing to address them promptly.
>
> Best regards, The Authors

---

### Official Review · Reviewer_wKx9 · 2024-08-10
**review for IMDL-BenCo**

**Rating:** 9
**Confidence:** 5
**Correctness:** Yes
**Clarity:** Yes

**Review:**

The paper presents a very significant contribution to the field of image forensics. The lack of standard evaluation protocol and efficient coding pipeline is the key challenge in IMDL field, and leads to many misleading claims or results. This paper introduces IMDL-BenCo, a unified benchmark and modular codebase designed to standardize and enhance the process of detecting and localizing image manipulations. The paper identifies and addresses key challenges within the IMDL domain through building the first comprehensive evaluation framework and many open-source baseline models. Besides, I do appreciate that this work also conducts 4 in-depth analysis and answered one key question I always concern: are the low feature extractors a must in IMDL. In general the paper's contribution to the field of image manipulation detection and localization is substantial. It offers a robust platform that will likely become a standard in the field, facilitating future research and development. The benchmark and codebase will be invaluable to both researchers and practitioners, driving progress in the detection and localization of image manipulations. Therefore, I suggest to accept this work.

**Strengths:**

1. modular approach of programming pipeline, which allows for greater flexibility and efficiency in model construction
2. The authors have also made a significant contribution by implementing a comprehensive set of state-of-the-art models within their benchmark.
3. the paper provides an in-depth analysis of various components critical to IMDL, such as the role of low-level feature extractors and the selection of appropriate backbone architectures. The discussion on dataset bias and the proposal of cleansing methods are particularly insightful, offering practical solutions to enhance the reliability of IMDL models.

**Additional Feedback:**

I have two questions for the authors:
1. Through table 6, I found most backbones are not benefit from the low level feature extractors. But in most works, introducing these extractors seems to be beneficial to the models. Is this own to the difference on evaluation protocols? Or can you claim that these exactors are overfitting to the dataset bias?
2. Will you release NIST16-C as well? with your findings, I believe cleaning NIST is an emergency for IMDL.

**Documentation:**

Yes

**Limitations:**

see "opportunities for improvement"

**Opportunities For Improvement:**

I have 2 suggestions for this work:
1. establish a schedule for your maintain and update on this codebase, especially the timetable to release your userguidelines and adding more (even all) SOTA models to your model zoo. Such that the research community will be informed and get ready to use your benchmark.
2. the F1 computation in 5.3. I understand the various selection of F1 computation is one of the factor affecting fair evaluation, so you examined these F1 scores and conclude to use "binary F1". You can offer the equation for various F1, and explain the differences more mathematically by comparing the equation differences. This will be more solid and straightforward.

**Relation To Prior Work:**

Yes

**Summary And Contributions:**

The paper presents a very significant contribution to the field of image forensics. The lack of standard evaluation protocol and efficient coding pipeline is the key challenge in IMDL field, and leads to many misleading claims or results. This paper introduces IMDL-BenCo, a unified benchmark and modular codebase designed to standardize and enhance the process of detecting and localizing image manipulations. The paper identifies and addresses key challenges within the IMDL domain through building the first comprehensive evaluation framework and many open-source baseline models. Besides, I do appreciate that this work also conducts 4 in-depth analysis and answered one key question I always concern: are the low feature extractors a must in IMDL. In general the paper's contribution to the field of image manipulation detection and localization is substantial. It offers a robust platform that will likely become a standard in the field, facilitating future research and development. The benchmark and codebase will be invaluable to both researchers and practitioners, driving progress in the detection and localization of image manipulations. Therefore, I suggest to accept this work.

---

> ### Author Rebuttal · Authors · 2024-08-11
>
> Thank you for your valuable feedback and thoughtful comments on our manuscript. We appreciate the time and effort you have invested in reviewing our work. Here is our response to address your concerns.
>
> ## R1-1 Various F1
> - Regarding macro-F1 and micro-F1, we will discuss their algorithms here:
> sklearn has designed these two metrics for calculating "multi-class F1" rather than binary F1. This means that F1 scores are computed for each class and then averaged. Specifically, when used for binary classification, an additional F1 score is calculated by reversing both the predicted mask and the ground truth (GT), and this score is averaged with the original F1 score. The specific averaging algorithms are different between macro and micro, and the mathematical formulas are as follows:
>
> - Macro-F1
>
> $$
> \begin{align}
> F1 &= 2\times \frac{TP}{2\times TP+FP+FN} \\\\
> F1'&= 2\times \frac{TN}{2\times TN+FN+FP} \\\\
> F1_{macro} &= \frac{F1 + F1'}{2} = \frac{TP}{2\times TP+FP+FN} + \frac{TN}{2\times TN+FN+FP} \\\\
> F1-F1{macro} &= \frac{TP}{2\times TP + FP + FN} - \frac{TN}{2\times TN +FN + FP} \\\\
> &= \frac{1}{2+\frac{FP+FN}{TP}} - \frac{1}{2+\frac{FN+FD}{TN}}
> \end{align}
> $$
> - Micro-F1
>
> $$
> \begin{align}
> F1 &= 2\times \frac{TP}{2\times TP+FP+FN} \\\\
> F1_{micro} &= \frac{TP+TN}{2\times(TP+TN) + 2\times(FP +FN)} \\\\
> F1-F1_{micro} &= 2\times \frac{TP}{2\times TP+FP+FN} - \frac{TP+TN}{2\times(TP+TN) + 2\times(FP +FN)} \\\\
> &= \frac{TP\times[2\times(TP+TN)+2\times(FP+FN)] - (TP+TN) \times(2\times TP + FP + FN)}{(2\times TP + FP + FN)[2\times (TP+TN) + 2\times(FP+FN)]} \\\\
> &= \frac{TP \times FP + TP \times FN - FP\times TN - FN \times{TN}}{(2\times TP + FP + FN)[2\times (TP+TN) + 2\times(FP+FN)]} \\\\
> &= \frac{(TP-TN)\times (FP + FN)}{(2\times TP + FP + FN)[2\times (TP+TN) + 2\times(FP+FN)]}
> \end{align}
> $$
> Since in IMDL the majority of cases typically have $TP << TN$, it follows that $F1 - F1_{macro} < 0$ and $F1 - F1_{micro} < 0$. Consequently, $F1 < F1_{macro}$ and $F1 < F1_{micro}$, leading to an erroneous estimation of model performance.
>
> We have also provided code to verify the specific implementation of macro-F1 and micro-F1 here: [Github script](https://github.com/scu-zjz/IMDLBenCo/blob/main/tests/test_sklearn_F1s.py)
>
> ## R1-2: Release NIST16-C
> - We have released it on this [GitHub repo](https://github.com/DSLJDI/NIST16-data-set-deduplication).
>
> ******
> Due to the formula having cost many characters, we will answer the other questions with comments.

---

> > ### Author Rebuttal · Authors · 2024-08-11
> >
> > ## R1-3: A schedule for your maintenance and update
> > Regarding this part, we have conducted preliminary planning discussions in the Supplementary Materials. For a detailed plan, we divided them into two parts:
> > - The completion of the documentation will be focused on after the Rebuttal period, with both Chinese and English versions being developed to ensure that individuals with a background in deep learning can easily start using it. We also welcome the community to translate it into other languages.
> > - Due to limited personnel, we can't add all SoTA models. We will only consider adding models with a good reputation in the industry in a timely manner. Of course, we also encourage future authors to develop based on IMDL-BenCo, as this will make merging and additions extremely easy and efficient through pull requests on GitHub, facilitating collaboration within the open-source community.
> >
> > ## R1-4: Issue of most backbones do not benefit from the low-level feature extractors
> > We tend to answer this question 2-fold:
> > - To some extent, this indeed indicates that simple feature extractors may not adapt well to performance. The authors believe this may be related to the fact that common resize operations effectively disrupt low-level features.
> > - This paper only tests **shallow fusion** methods, specifically **concatenating the feature extractor with the original RGB image at the first layer**. This method is relatively more suitable for CNNs, as it allows for gradual fusion, whereas for transformers, it becomes a bottleneck because it imposes almost all the fusion tasks on the embedding layer.
> > - Of course, this can not be concluded with `Deeper fusion performance better`. The practice is the sole criterion for testing the truth. Thus, further experiments on it are needed. However, Due to limitations in length, this paper cannot fully explore this issue. (Like testing on various resolutions, explore deeper feature fusion methods.) Anyway, using the IMDL-Benco framework can efficiently conduct corresponding ablation experiments and related analyses for future study.

---

> ### Author Response · Authors · 2024-08-24
>
> Dear Reviewer wKx9,
>
> Thank you for your thorough evaluation of our work. We are committed to incorporating your feedback comprehensively into our revised paper to improve both its content and overall quality.
>
> As the discussion period draws to a close, we hope our response has sufficiently addressed your concerns. If there are any additional issues or points that require further clarification, we are more than willing to address them promptly.
>
> Best regards,
> The Authors

---

> > ### Comment · Reviewer_wKx9 · 2024-08-29
> >
> > The authors have well-addressed my two concerns, particularly regarding the computation of F1 score. These equations are vital in representing the variations among different version of F1 scores, so the author shall include these parts in their final version as they promised.
> > I checked their codebased along these days and confirmed that the author are conducting regular updating and maintain. This is a very good sign for a benchmark work.
> > I also read through other reviewers’ comments and find that they share similar positive attitudes toward this work as well.
> > So, considering above all, I decide to raise my score to 9 (since my original rate is already 8) to show my vote for this work.

---

> > ### Author Response · Authors · 2024-08-29
> >
> > Thank you for your valuable feedback and recognition of our work, especially regarding the F1 score calculation. We have included the relevant equations in the revised version as promised and will continue to maintain and update the codebase. We appreciate your support and the positive feedback as well.

---

### Decision · Program_Chairs · 2024-09-26

**Decision:**

Accept (Spotlight)

**Comment:**

This work built the first benchmark in the Image Manipulation Detection & Localization (IMDL) field, with modular codebase and extensive analysis.

It received 5 details reviews, with consistently positive scores. Most reviewers recognized the the value of this benchmark, due to the modular codebase, implementations of several sota methods, in-depth analysis with new findings. There are also several concerns and suggestions in the initial reviews, such as the implementation details. The authors addressed these concerns well in the rebuttal. It is required to add these additional evaluations, analysis and illustrations in the rebuttal into the final manuscript.

Considering the importance of the studied emerging topic, this work may appeal much attention from the community. Thus, it is recommended as oral.